# piRNA processing by a trimeric Schlafen-domain nuclease

Nadezda Podvalnaya[1,2,9], Alfred W. Bronkhorst[1,9], Raffael Lichtenberger[3,4], Svenja Hellmann[1], Emily Nischwitz[2,5], Torben Falk[3,4], Emil Karaulanov[6], Falk Butter[4,8], Sebastian Falk[3,4 ✉] & René F. Ketting[1,7 ✉]

Transposable elements are genomic parasites that expand within and spread between genomes[1]. PIWI proteins control transposon activity, notably in the germline[2,3]. These proteins recognize their targets through small RNA co-factors named PIWI-interacting RNAs (piRNAs), making piRNA biogenesis a key specificity-determining step in this crucial genome immunity system. Although the processing of piRNA precursors is an essential step in this process, many of the molecular details remain unclear. Here, we identify an endoribonuclease, precursor of 21U RNA 5′-end cleavage holoenzyme (PUCH), that initiates piRNA processing in the nematode *Caenorhabditis elegans*. Genetic and biochemical studies show that PUCH, a trimer of Schlafen-like-domain proteins (SLFL proteins), executes 5′-end piRNA precursor cleavage. PUCH-mediated processing strictly requires a 7-methyl-G cap (m⁷G-cap) and a uracil at position three. We also demonstrate how PUCH interacts with PETISCO, a complex that binds to piRNA precursors[4], and that this interaction enhances piRNA production in vivo. The identification of PUCH concludes the search for the 5′-end piRNA biogenesis factor in *C. elegans* and uncovers a type of RNA endonuclease formed by three SLFL proteins. Mammalian Schlafen (SLFN) genes have been associated with immunity[5], exposing a molecular link between immune responses in mammals and deeply conserved RNA-based mechanisms that control transposable elements.

Transposable elements are segments of DNA that can independently multiply and move within, and sometimes between genomes[1]. Being found in almost all genomes analysed to date, transposons are highly successful, and their control, especially in the germ cells, is an essential process. Notably, transposable elements can mutate to avoid defence systems and, in turn, defence systems can adapt to such changes, resulting in a molecular arms race that leads to rapid diversification between species[6]. Small-RNA-driven gene regulatory pathways represent one of the mechanisms through which transposable elements are controlled[2,3]. In animal germ cells, Argonaute proteins of the PIWI clade interact with piRNAs to control transposons, but also host genes[7]. This process is essential for germ cell function and fertility. piRNA pathways are characterized by many species-specific factors, even though piRNA pathways also share deeply conserved concepts[2,3].

The piRNA portfolio defines the target range and specificity of the PIWI–piRNA pathway. Mature piRNAs are generated from single-stranded piRNA precursor molecules[2,3,8]. This process is started by a nucleolytic cleavage, which defines the 5′-end of a new piRNA, which then is bound by a yet unloaded PIWI protein. In *Drosophila* and mouse, this cleavage can be executed by PIWI proteins themselves[2,3,8], leading to piRNA amplification, or by an endonuclease that goes by the names Zucchini (*Drosophila*, Zuc) or phospholipase D6

(mouse, PLD6)[9–12]. Zuc not only amplifies but also diversifies piRNA populations[13,14]. After 5′-end processing, it is believed that the 3′-end is processed after binding to a PIWI protein. This step involves trimming by 3′–5′ exoribonuclease activity and methylation of the 2′-OH at the 3′-end. In *C. elegans*, the trimming and methylation are done by PARN-1 (ref. 15) and HENN-1, respectively[16–18] (Fig. 1a).

Notably, not all animals rely on Zuc/PLD6 and/or PIWI for piRNA biogenesis. For instance, *C. elegans* lacks a Zuc homologue. Furthermore, the slicer activity of the *C. elegans* PIWI homologue (PRG-1)[19–21] is not needed for piRNA production[22], making it unclear how piRNA 5′-ends are generated. In this nematode, two types of piRNAs are found. Type 1 piRNA precursors are transcribed from short genes, each encoding one piRNA, which, in *C. elegans*, is also named 21U RNA[23]. The precursors of this most abundant class of piRNAs are around 27–29 nucleotides long and carry a 5′-cap[24] (Fig. 1a). In contrast to many other animals, including mammals, most *C. elegans* mRNAs do not have m⁷G-caps, but 2,2,7-trimethyl-G (TMG) caps through a process of 5′-end *trans*-splicing[25], and it is possible that m⁷G-caps help to distinguish between piRNA precursors and mRNAs. Indeed, non-*trans*-spliced, but capped short transcripts from certain genes can be processed into piRNAs. These are much less abundant, and known as type 2 piRNAs[24]. After transcription by specialized machinery[26,27], piRNA precursors are

[1]Biology of Non-coding RNA group, Institute of Molecular Biology, Mainz, Germany. [2]International PhD Programme on Gene Regulation, Epigenetics & Genome Stability, Mainz, Germany. [3]Max Perutz Labs, Vienna Biocenter Campus (VBC), Vienna, Austria. [4]Department of Structural and Computational Biology, Center for Molecular Biology, University of Vienna, Vienna, Austria. [5]Quantitative Proteomics group, Institute of Molecular Biology, Mainz, Germany. [6]Bioinformatics Core Facility, Institute of Molecular Biology, Mainz, Germany. [7]Institute of Developmental Biology and Neurobiology, Johannes Gutenberg University, Mainz, Germany. [8]Present address: Institute of Molecular Virology and Cell Biology, Friedrich Loeffler Institute, Greifswald, Germany. [9]These authors contributed equally: Nadezda Podvalnaya, Alfred W. Bronkhorst. ✉e-mail: sebastian.falk@univie.ac.at; r.ketting@imb-mainz.de

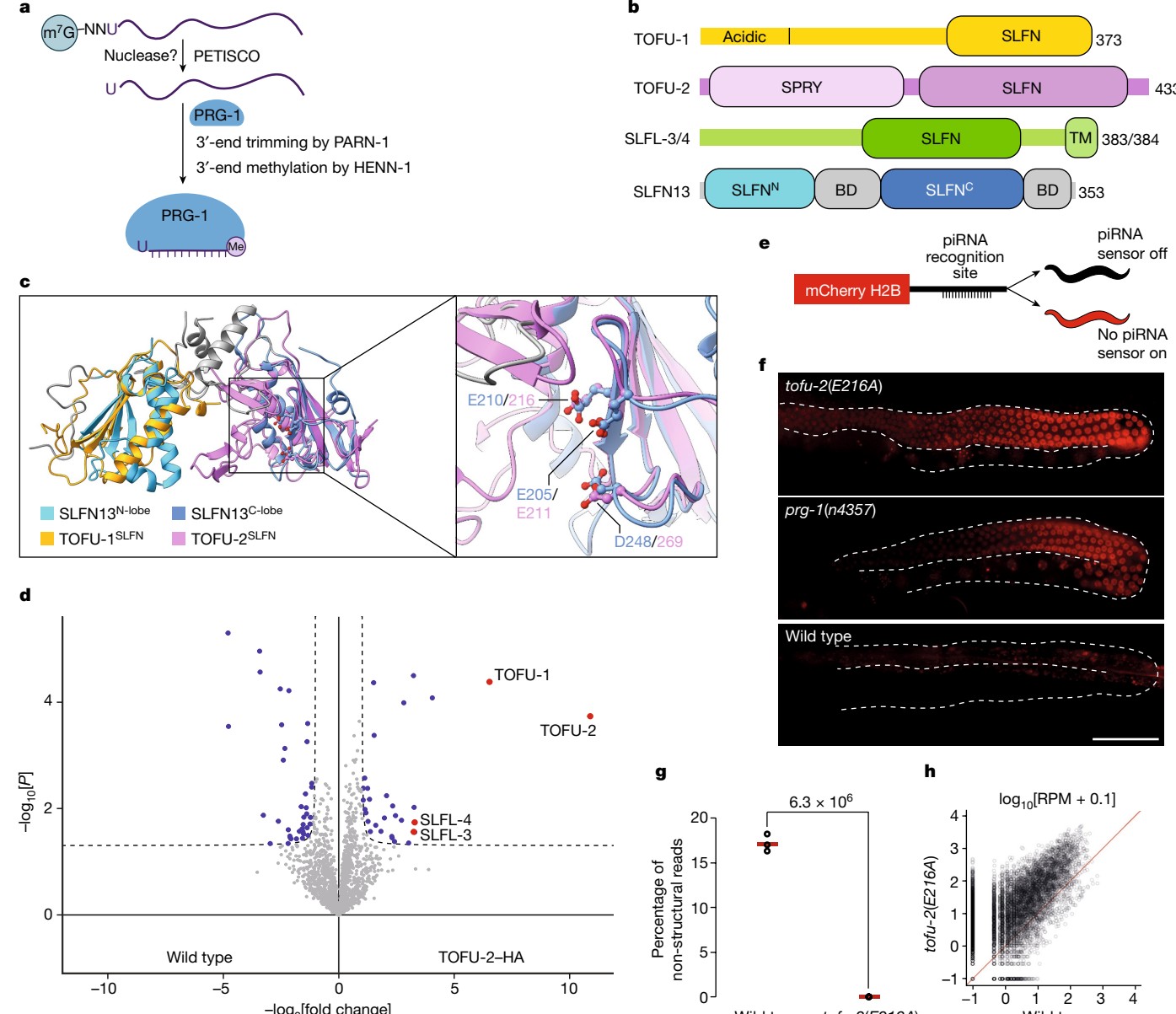

**Fig. 1 | Identification of the catalytic centre of TOFU-2. a**, Model of piRNA (21U RNA) formation in *C. elegans*. Individually transcribed piRNA precursors are stabilized by PETISCO. After the removal of the 5′-cap and two nucleotides, intermediates are loaded onto PRG-1, followed by trimming and 3′-end methylation. The nuclease that processes the 5′-end is currently unclear. **b**, Schematic of TOFU-1, TOFU-2 and SLFL-3/4, in comparison to rat SLFN13. The lines indicate low-complexity regions and the rectangles indicate the predicted folded domains. BD, bridging domain. **c**, Superposition of TOFU-1 and TOFU-2 SLFN domains onto the crystal structure of the N-terminal SLFN13 endoribonuclease domain (Protein Data Bank (PDB): 5YD0). Domains are coloured as in **b**. The magnified view shows the active site of SLFN13. Involved residues are shown as sticks. **d**, Label-free proteomic quantification of TOFU-2–HA and wild-type immunoprecipitates from young adult extracts. *n* = 4 biological replicates. The *x* axis shows the median fold enrichment of individual proteins, and the *y* axis shows −log$_{10}$[*P*]. *P* values were calculated using Welch two-sided

*t*-tests. The dashed lines represent enrichment thresholds at *P* = 0.05 and fold change > 2, curvature of enrichment threshold *c* = 0.05. The dots represent enriched (blue/red) or quantified (grey) proteins. Only uniquely matching peptides were used. **e**, Schematic of the mCherry–H2B piRNA sensor. **f**, Widefield fluorescence microscopy analysis of adult hermaphrodites carrying the piRNA sensor in the following three genetic backgrounds: *tofu-2(E216A)* (top), *prg-1(n4357)* (middle) and wild type (bottom). Germlines are outlined by white dashed lines. Scale bar, 50 μm. A representative image from a series of ten is shown. **g**, Total mature piRNA levels (type 1) in wild-type and *tofu-2(E216A)*-mutant young adult hermaphrodites. *n* = 3 biological replicates. The red lines show the group means. *P* values were calculated using two-tailed unpaired *t*-tests. **h**, The relative abundance of type 1 piRNA precursors from individual loci in *tofu-2(E216A)*-mutant versus wild-type young adult hermaphrodites. *n* = 3 biological replicates. RPM, reads per million non-structural small RNA reads.

bound by PETISCO—a cytoplasmic protein complex consisting of PID-3, ERH-2, TOFU-6 and IFE-3 (refs. 4,28–30)—followed by the removal of the m$^7$G-cap together with the first two nucleotides. Although PETISCO has been implicated in precursor stabilization and is required for piRNA production, it contains no nucleases. Thus, the nuclease that mediates 5′ precursor processing and how it interacts with PETISCO remain unclear.

## The TOFU-1–TOFU-2 complex is a potential nuclease

A genome-wide RNA interference screen identified the proteins TOFU-1 and TOFU-2 as factors that are necessary for piRNA accumulation[31]. The loss of these factors also triggered piRNA precursor accumulation, suggesting they may have a role in piRNA 5′-end processing. However,

domain annotations at that time did not reveal potential nuclease domains. Using structure-based homology searches (HHPRED) and AlphaFold2, we detected homology between TOFU-1 and TOFU-2, and the rat ribonuclease SLFN13 (ref. 32), but also human SLFN5 (ref. 33) and SLFN12 (ref. 34). This identified the presence of a potential SLFN-fold in both TOFU-1 and TOFU-2 (Fig. 1b,c). Notably, whereas two SLFN-folds come together to form the nuclease domain in mammalian Schlafen proteins[32–34], in TOFU-1 and TOFU-2, only one SLFN-fold could be identified. We therefore hypothesized that TOFU-1 and TOFU-2 may interact to form a functional nuclease. To test this hypothesis, we tagged endogenous TOFU-2 with a human influenza haemagglutinin (HA) tag and used immunoprecipitation followed by quantitative mass spectrometry (IP–MS) to identify TOFU-2-interacting proteins. Indeed, TOFU-1 was found to interact with TOFU-2 (Fig. 1d and Supplementary Table 1). Potential catalytic residues were identified within TOFU-2, but not within TOFU-1 (Fig. 1c and Extended Data Fig. 1a). We therefore engineered a *C. elegans tofu-2* mutant in which we changed one of the potential catalytic residues (glutamic acid 216) to alanine (*tofu-2(E216A)*). This mutation neither affects TOFU-2 abundance nor interaction with TOFU-1, as determined using western blotting and IP–MS analysis (Extended Data Fig. 1b,c and Supplementary Table 2). We next tested the piRNA silencing activity in this mutant using a piRNA sensor (a germline-expressed transgene that is silenced through piRNA activity[22]) (Fig. 1e). This revealed that *tofu-2(E216A)* mutants de-silence the piRNA sensor to a similar extent as *prg-1* mutants (Fig. 1f). Sequencing of piRNAs and piRNA precursors showed that *tofu-2(E216A)* mutants lost almost all mature piRNAs and accumulated precursors (Fig. 1g,h and Extended Data Fig. 1d,e). We conclude that a TOFU-1–TOFU-2 complex could be the nuclease that processes piRNA precursors.

## SLFL-3 or SLFL-4 binds to TOFU-1–TOFU-2

Next, we heterologously expressed TOFU-1 and TOFU-2 in BmN4 cells, a cell culture system derived from the silk moth ovary that expressed these proteins well. We found that TOFU-1 and TOFU-2 co-immunoprecipitate and that TOFU-1 stabilizes TOFU-2 (Fig. 2a (lanes 4 and 6)). However, incubating the co-immunoprecipitates with a synthetic piRNA precursor did not result in precursor cleavage (see the next section), suggesting that our experimental conditions might lack an essential co-factor.

The TOFU-2 IP–MS experiments, in addition to TOFU-1, also identified the proteins C35E7.8 and F36H12.2 (Fig. 1d and Extended Data Fig. 1a,c). These two proteins are 90% identical at the amino acid level (Extended Data Fig. 2a) and may therefore function redundantly. Analysis using AlphaFold2 revealed that these two proteins also contain a single potential SLFN-like fold (Extended Data Figs. 1f and 2b). We therefore propose the name SLFN-like, or SLFL, for this group of proteins that contain only a single SLFN-fold, with TOFU-1, TOFU-2, C35E7.8 and F36H12.2 corresponding to SLFL-1, SLFL-2, SLFL-3 and SLFL-4, respectively. SLFL-3 was identified in the same study that identified TOFU-1 and TOFU-2, but its RNA-interference-mediated knockdown triggered a relatively weak reduction in piRNA levels and was not investigated further[31]. We generated a *slfl-3* deletion mutant and found that this allele triggers mild activation of the piRNA sensor (Extended Data Fig. 2c). However, this activation was lost in later generations. This resembles what we previously observed in *henn-1* mutants, in which the piRNA pathway is crippled, but not inactivated[35]. To more rigorously examine the involvement of SLFL-3 and SLFL-4, we also generated a *slfl-4*-deletion mutant and sequenced piRNAs from single and double mutants. The *slfl-3;slfl-4* double mutants almost completely lost piRNAs (Fig. 2b and Extended Data Fig. 2d), and displayed precursor accumulation (Fig. 2c), whereas the single mutants did not show defects (Fig. 2b and Extended Data Fig. 2e). We conclude that SLFL-3 and SLFL-4 function redundantly in piRNA processing.

In addition to the SLFN-like fold, SLFL-3 and SLFL-4 also contain a predicted transmembrane (TM) helix (Extended Data Fig. 2a,b,f), a feature that is also present in mammalian and *Drosophila* Zuc[36,37]. By transfecting BmN4 cells with TOFU-1, eGFP-TOFU-2 and mCherry-SLFL-3 carrying or lacking the TM helix, we showed that the TM helix mediates localization to mitochondria (Fig. 2d). Notably, TOFU-2 mirrored SLFL-3 localization, suggesting that they form a complex. To examine the relevance of the TM helix of SLFL-3, we generated a TM-helix-deletion allele of *slfl-3* (*slfl-3(ΔTM)*), crossed it into a *slfl-4*-mutant background and analysed piRNA levels by sequencing. This revealed a strong reduction in mature piRNAs (Fig. 2b and Extended Data Fig. 2d), suggesting that mitochondrial proximity is important for piRNA production. Notably, piRNA precursor levels were unaffected (Extended Data Fig. 2e).

We used AlphaFold2 to predict how these four SLFL proteins may interact with each other. This revealed that a trimeric combination of TOFU-1, TOFU-2 and either SLFL-3 or SLFL-4 yielded the best predictions, in which the three SLFN domains were found to interact with each other (Fig. 2e and Extended Data Fig. 3). Further fine-tuning the procedure produced a high-confidence model of TOFU-1, TOFU-2 and SLFL-3 (Fig. 2e and Extended Data Fig. 4) suggesting that the active nuclease may be a trimeric complex. This prompted us to co-express TOFU-1, TOFU-2 and either SLFL-3 or SLFL-4 in BmN4 cells and to test their interaction using co-immunoprecipitation experiments. Indeed, these experiments support the idea of a trimer. For example, SLFL-3 further enhances TOFU-2 expression, but only in the presence of TOFU-1 (Fig. 2a (lanes 3–5)). Also, in absence of TOFU-1, we could not detect interactions between TOFU-2 and SLFL-3 (Fig. 2a (lane 5)).

We also assessed the interactions between TOFU-1, TOFU-2 and SLFL-3 through heterologous expression and co-immunoprecipitation experiments in *Escherichia coli* (Extended Data Fig. 5a). While the TOFU-2 SPRY domain did not display strong interactions (Extended Data Fig. 5b), TOFU-1 and SLFL-3 interacted through their SLFN domain directly with the TOFU-2 SLFN domain (Extended Data Fig. 5c–e), and a complex containing all three proteins could be readily identified (Extended Data Fig. 5e–g). These findings are consistent with the AlphaFold2 model (Extended Data Fig. 4) and the co-immunoprecipitation analysis in BmN4 cells (Fig. 2a).

## PUCH is a trimeric piRNA precursor nuclease

We next tested co-immunoprecipitates from BmN4 cells, in which we co-expressed different combinations of TOFU-1, TOFU-2, SLFL-3 and SLFL-4, for piRNA-processing activity. As a substrate, we used a synthetic piRNA precursor oligonucleotide carrying an m[7]G-cap, which was radioactively labelled at its 3′-end with $^{32}$P for detection (Fig. 3a). Processing activity was analysed on a denaturing polyacrylamide gel system, alongside a synthetic RNA representing the expected processing product. This yielded processing activity, but only when both TOFU-1 and TOFU-2, as well as either SLFL-3 or SLFL-4, were present (Fig. 3b). Introduction of an E216A mutation into TOFU-2 completely blocked this cleavage reaction (Fig. 3b). Mammalian SLFN nucleases require divalent cations for cleavage activity[32,38]. Likewise, precursor processing was inhibited by EDTA, and was supported by divalent cations such as $Mg^{2+}$, $Mn^{2+}$ or $Ca^{2+}$ (at high concentrations), but not by $Zn^{2+}$ (Extended Data Fig. 6a,b).

To exclude the possibility that any BmN4-derived factors were responsible for the cleavage reaction, we also expressed both active and inactive (E216A) minimal versions of the TOFU-1–TOFU-2–SLFL-3 complex recombinantly in *E. coli* (Extended Data Fig. 5f–h). This minimal complex was active in precursor cleavage assays, while the E216A mutant was not (Fig. 3c).

Mature piRNAs carry a monophosphate at their 5′-ends, which would be consistent with the cleavage product of a metal-dependent nuclease[32,39]. Successful ligation of the cleavage product to a synthetic RNA oligonucleotide with hydroxyl groups at both 5′ and 3′-ends confirmed the presence of a 5′-phosphate (P) on the reaction product of the TOFU-1–TOFU-2–SLFL-3 nuclease (Fig. 3d). On the basis of these

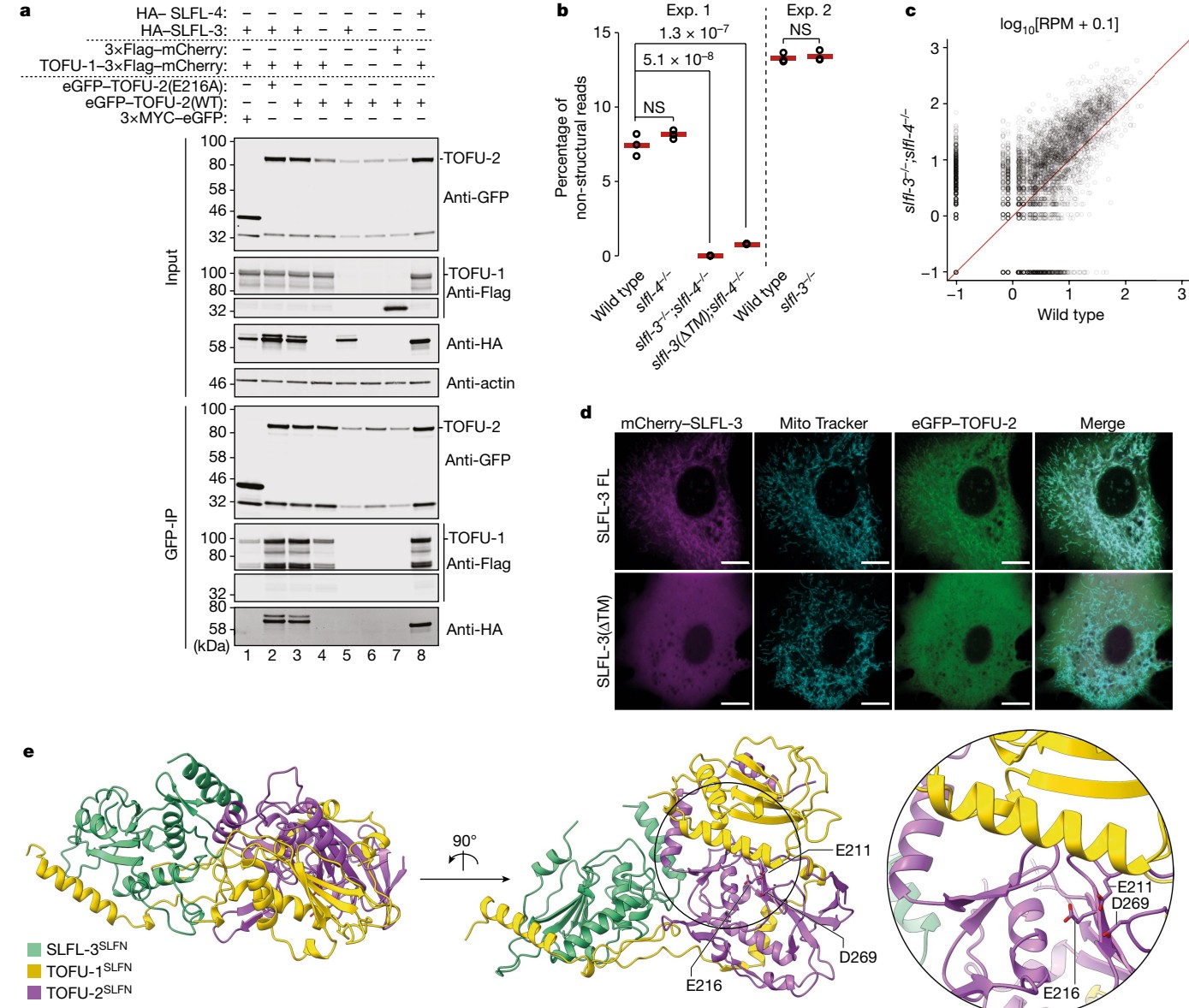

**Fig. 2 | TOFU-1, TOFU-2 and SLFL-3/4 form a mitochondria-bound complex.**
**a**, Anti-GFP immunoprecipitation analysis of BmN4 cell lysates of cells
that were transfected with the indicated constructs. eGFP–TOFU-2 was
immunoprecipitated, followed by western blot detection of TOFU-1 (Flag),
SLFL-3 (HA) or SLFL-4 (HA). Expression of 3×MYC–eGFP and of 3×Flag–mCherry
served as negative controls. Note that low TOFU-2 levels in lanes 5 and 7 may
limit the detection of interactions. All observations were performed at least in
duplicate. IP, immunoprecipitation. **b**, Total mature piRNA levels (type 1) in
young adult hermaphrodites of the indicated genotypes. $n = 3$ biological
replicates. The red lines depict group means and *P* values were calculated using
one-way analysis of variance (ANOVA) followed by Tukey's honest significant
difference (HSD) test (left) and two-tailed unpaired *t*-tests (right). The plot is
based on two independent experiments (exp. 1 and 2). NS, non-significant.

**c**, The relative abundance of type 1 piRNA precursors from individual loci in
*slfl-3*[−/−];*slfl-4*[−/−] mutant versus wild-type young adult hermaphrodites. $n = 3$
biological replicates. **d**, Single-plane confocal micrographs of BmN4 cells that
were transfected with eGFP–TOFU-2 and full-length mCherry–SLFL-3 (top) or
mCherry–SLFL-3(ΔTM) (bottom). TOFU-1 was also transfected but was not
tagged with a fluorescent protein. Mitochondria were stained with Mito
Tracker. Scale bars, 10 µm. The experiment was performed in duplicate; a
representative image from a series of 20 is shown. **e**, AlphaFold2-predicted
structure of a minimal trimeric TOFU-1–TOFU-2–SLFL-3 complex. The best of
five predicted models is shown as a cartoon in two different orientations.
TOFU-1 is shown in yellow, TOFU-2 in purple and SLFL-3 in green. The TOFU-2
active-site residues are shown as a stick representation and are magnified at
the bottom right. Raw data are provided in Supplementary Fig. 1.

results, we conclude that a complex of TOFU-1, TOFU-2 and either SLFL-3
or SLFL-4 constitutes the enzyme that processes the 5′-end of piRNA
precursors in *C. elegans*. We name this complex PUCH.

## PUCH acts cap and sequence specifically

We probed key piRNA precursor properties for their relevance to pro-
cessing. First, piRNA precursors are characterized by a 5′-m⁷G-cap[24].
To examine whether the cap structure is essential for PUCH activity,

we incubated full-length PUCH (isolated by TOFU-2 immunoprecipi-
tation from BmN4 cell extracts) with a precursor with a 5′-P instead of
a 5′-m⁷G-cap. This experiment revealed that 5′-P precursor RNA was
not processed, in contrast to the capped control substrate (Fig. 3e).
A second piRNA-precursor characteristic in *C. elegans* is the presence
of a uracil at position three (U3)[24]. This corresponds to the most 5′
nucleotide in mature piRNAs, which display an extreme 5′-U bias[23].
We tested whether PUCH could process a precursor substrate contain-
ing a cytosine at position three (AAC precursor) and found that PUCH

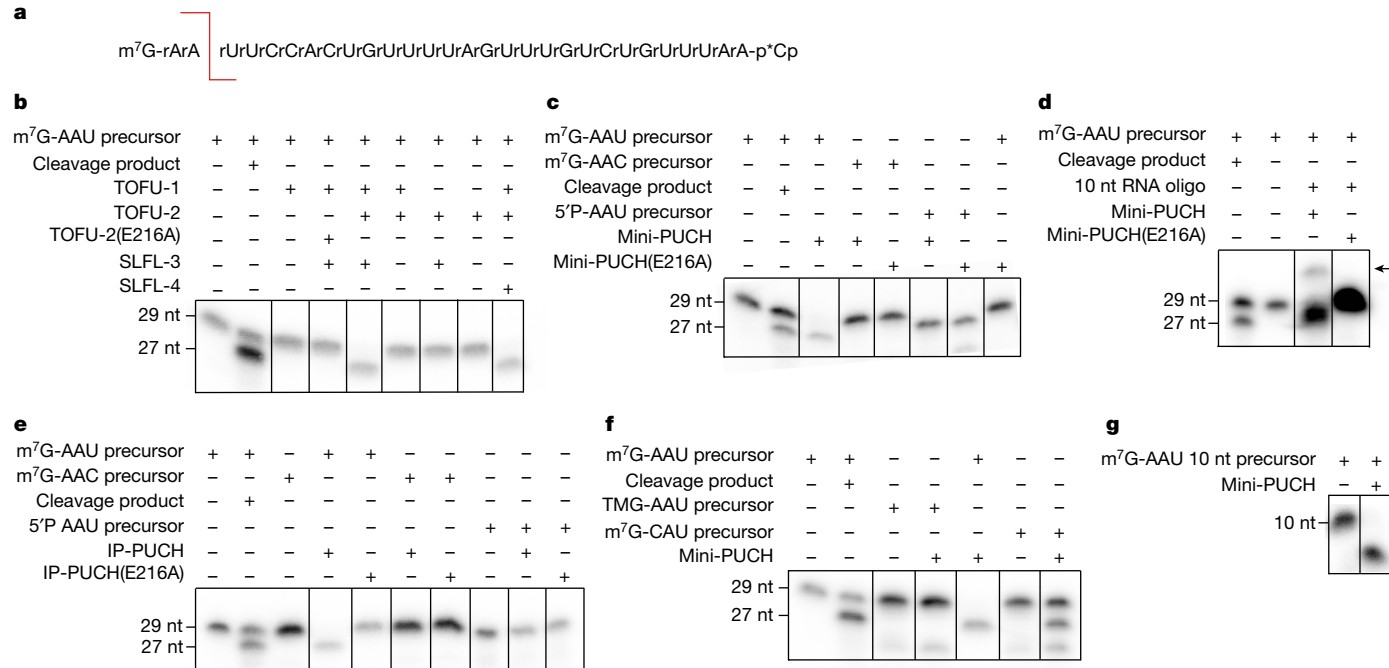

**Fig. 3 | PUCH is a cap- and sequence-specific endoribonuclease. a**, The sequence of the synthetic piRNA precursor used in the assay. The red line indicates the expected cleavage position. Both the precursor and expected cleavage product were run in the two left-most lanes of every gel to mark where these molecules are expected. **b**, In vitro cleavage assay of the piRNA precursor using anti-GFP immunoprecipitated material from BmN4 cell extracts. Cells were transfected with eGFP–TOFU-2, TOFU-1, SLFL-3 or SLFL-4 at various combinations, as indicated. All observations were performed at least in duplicate. nt, nucleotides. **c**, Cleavage assays with recombinant minimal PUCH (mini-PUCH) and different RNA substrates. E216A indicates the presence of

TOFU-2 containing the catalytic E216A mutation. All observations were performed at least in duplicate. **d**, RNA obtained from a cleavage reaction (using either wild-type or TOFU-2(E216A)-mutant mini-PUCH) was ligated to a 10-nucleotide-long 5'OH-containing RNA adapter. The ligation product is indicated by an arrow. The experiment was performed in triplicate. **e**, In vitro cleavage assay on different types of RNA substrate using the PUCH complex retrieved from BmN4 cells by immunoprecipitation (IP). All observations were performed at least in duplicate. **f,g**, Cleavage assays with mini-PUCH and the indicated substrates. The experiment was performed in triplicate for **f** and once for **g**. Raw data are provided in Supplementary Fig. 1.

did not cleave the AAC precursor at detectable levels (Fig. 3e). A third characteristic of precursors is a strong bias for an A or G at position 1. To investigate its relevance for processing, we tested a substrate in which we changed the first nucleotide to a C. This CAU substrate was cleaved, but more slowly than the AAU substrate (Fig. 3f and Extended Data Fig. 6c–e). Similar results were obtained with the recombinant minimal PUCH complex, containing only the SLFN domains of the three subunits (mini-PUCH) (Fig. 3c,f). Using mini-PUCH we also demonstrated that a TMG-cap prevented cleavage (Fig. 3f). Finally, shortening of the substrate at the 3'-end did not affect cleavage (Fig. 3g), and none of the cleavage-incompetent substrates inhibited processing of the canonical AAU substrate (Extended Data Fig. 6f). We conclude that PUCH is a type of cap- and sequence-specific ribonuclease.

## PUCH cleaves PETISCO-bound precursors

In vivo, piRNA precursors are bound by PETISCO[4,28], and this enhances piRNA biogenesis. Yet, based on the results described thus far, PUCH does not require PETISCO for activity in vitro. PETISCO's main role may therefore be to stabilize precursors in vivo, and not to promote PUCH activity. To genetically probe the relationship between PETISCO and PUCH, we examined how the loss of PETISCO function affects precursor accumulation in *tofu-2(E216A)* mutants. To this end, we sequenced small RNAs from a strain carrying the *tofu-2(E216A)* allele and lacking the piRNA-specific PETISCO adapter protein PID-1 (ref. 40). In *tofu-2(E216A);pid-1(xf35)* double mutants, precursor accumulation was reduced (Fig. 4a), consistent with the idea that PETISCO stabilizes piRNA precursors to allow their processing by PUCH. Mature piRNAs were completely absent, as in *tofu-2(E216A)* single mutants (Fig. 4b).

These results also imply that PUCH can process piRNA precursors while they are bound by PETISCO. To test this directly, we first incubated [32]P-labelled precursors with purified PETISCO and tested binding in an electromobility-shift assay (EMSA). We observed that the substrate was indeed bound by PETISCO, resulting in most of the complex not being able to enter the gel, most likely due to the large size of PETISCO (octameric complex of 240 kDa)[28]. The presence of a 5'-m[7]G-cap on the precursor enhanced RNA binding by PETISCO (Extended Data Fig. 6g). We next incubated PETISCO-precursor complexes with full-length immunopurified PUCH, or recombinant minimal-PUCH and analysed the cleavage products in a time series. This revealed that cleavage is not prevented by the presence of PETISCO (Fig. 4c,d and Extended Data Fig. 6h,i). We conclude that PUCH can cleave piRNA precursors, also in the presence of PETISCO.

## PUCH–PETISCO interaction

If PUCH cleaves PETISCO-bound precursors, interactions between the two complexes may be expected. However, multiple IP–MS experiments, including co-immunoprecipitation of TOFU-2(E216A), did not reveal interactions between PUCH subunits and PETISCO (Extended Data Fig. 1d and Supplementary Tables 1 and 2). Reasoning that the presumed interaction may be too transient to be detected in *C. elegans* extracts, we systematically tested interactions between recombinant proteins in pull-down assays. This revealed an interaction between TOFU-1 and PETISCO (Fig. 5a). Using a combination of pull-down and size-exclusion chromatography experiments, we narrowed down the interaction to a region upstream of the TOFU-1 SLFN domain (residues 82–172, TOFU-1[N]) and to the extended TUDOR (eTUDOR) domain of the PETISCO subunit

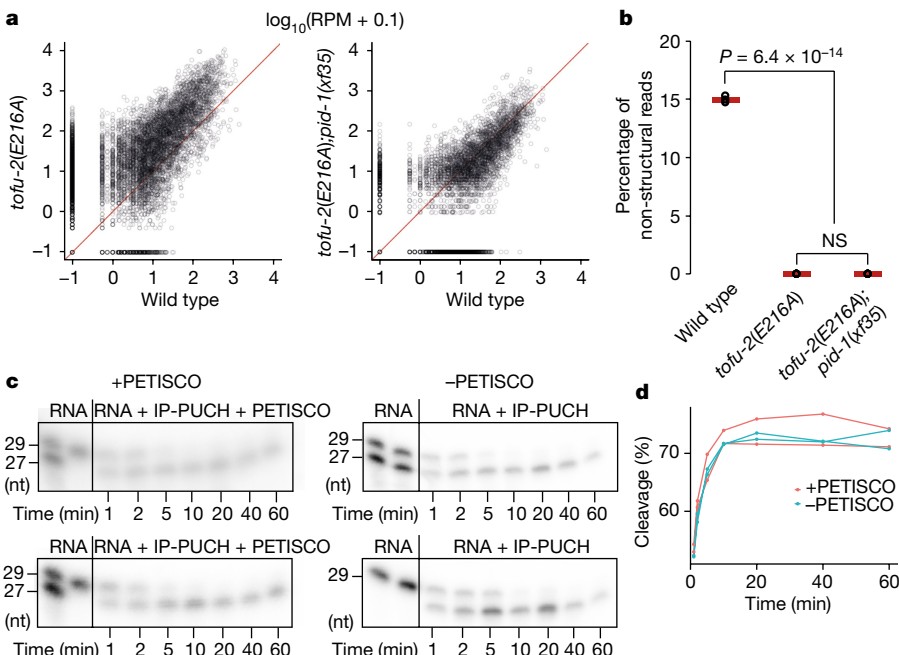

**Fig. 4 | PETISCO is necessary for piRNA precursor accumulation in vivo and does not interfere with PUCH-mediated precursor cleavage. a**, The relative abundance of individual type 1 piRNA precursors in *tofu-2(E216A)* mutant (left) and *tofu-2(E216A);pid-1(xf35)* double-mutant (right) versus wild-type young adult hermaphrodites. *n* = 3 biological replicates. **b**, Total mature piRNA levels (type 1) in wild-type, *tofu-2(E216A)* mutant and *tofu-2(E216A);pid-1(xf35)* double-mutant young adult hermaphrodites. *n* = 3 biological replicates. The red lines show the group means. *P* values were calculated using one-way ANOVA followed by Tukey's HSD test; the indicated *P* value relates to both mutant samples. NS, non-significant. **c**, In vitro piRNA precursor cleavage assays in the presence or absence of the PETISCO complex in a time series, in duplicate. In these experiments, PUCH was isolated from BmN4 cell extracts by immunoprecipitation (IP). **d**, Quantification of the cleavage reactions presented in **c**. Raw data are provided in Supplementary Fig. 1.

TOFU-6 (TOFU-6^eTUDOR) (Extended Data Fig. 7a–e). AlphaFold predictions showed two helices in the N-terminal part of TOFU-1, and a construct containing only the first helix (residues 82–113, TOFU-1^pep) was sufficient to bind to TOFU-6^eTUDOR (Fig. 5b and Extended Data Fig. 8a). Quantitative analysis with the minimal TOFU-1^pep using isothermal titration calorimetry revealed a $K_d$ of around 20 μM (Extended Data Fig. 7f). We determined the crystal structure of the TOFU-6^eTUDOR–TOFU-1^pep complex at a resolution of 2.2 Å (Fig. 5c and Extended Data Table 1). TOFU-1^pep does not bind to TOFU-6^eTUDOR at the canonical, dimethyl-arginine-binding aromatic cage of the TUDOR domain[41,42], but on the surface of the staphylococcal nuclease-like domain of the eTUDOR domain (Fig. 5c and Extended Data Fig. 9a–c). To date, this region has not been described to mediate protein–protein interactions to our knowledge. On the basis of the interaction interface, we designed mutations in both TOFU-1^pep and TOFU-6^eTUDOR that should disrupt their interaction and tested these using pull-downs and size-exclusion chromatography (Extended Data Fig. 8a–e). Whereas mutations on only one of the partners (especially TOFU-1^pep) weakened the interaction, mutation of both partners fully disrupted the interaction. We next tested the same mutations in vivo, using CRISPR–Cas9-mediated mutagenesis of the endogenous loci. The introduced mutations did not affect protein abundance (Fig. 5d). Sequencing of piRNAs and their precursors from both single and double mutants revealed a reduction in mature piRNAs, as well as an accumulation of precursors (Fig. 5e–g). We conclude that piRNA accumulation in vivo is stimulated by the interaction between PETISCO and PUCH.

## Discussion

The identification of PUCH represents an important expansion of the piRNA biogenesis toolkit of *C. elegans*. At the sequence level, PUCH is unrelated to Zuc, the enzyme that initiates piRNA biogenesis in mammals and flies. Yet, both enzymes perform a similar reaction: they both cleave piRNA precursors at a specified distance from the 5′-end of the precursor. Whereas Zuc depends on PIWI proteins binding to precursor 5′ ends[11,13,14], PUCH depends on a 5′-m^7G-cap, which is probably bound by PUCH itself. This specificity may contribute to substrate selection and help to safeguard *trans*-spliced mRNAs from the PETISCO–PUCH machinery. If and how PUCH is prevented from cleaving non-*trans*-spliced transcripts other than piRNA precursors is a subject of future study. Type 2 piRNAs[24] may reflect accidental targeting of non-*trans*-spliced transcripts by PUCH. Whether type 2 piRNAs indeed represent off-target substrates of the pathway or whether they have a function is currently unclear. A second commonality between Zuc and PUCH is the requirement of a uracil downstream of the cleaved phosphodiester bond. While for Zuc this is a rather weak requirement[11], for PUCH, this is a prerequisite for cleavage. This imposes a strong selection on potential new sequences that may evolve towards piRNA precursors. A third similarity between the enzymes is that both contain a TM helix. Zuc is bound to the mitochondrial outer membrane through an N-terminal TM helix[36,37], whereas PUCH is brought to mitochondria through a C-terminal TM helix on SLFL-3/4. Deletion of the TM helix results in a strong reduction in mature piRNAs without precursor accumulation, which is different to the *tofu-2* and *slfl-3/4* phenotypes. This may indicate that precursor processing per se is not affected. Possibly, the loading of processing intermediates into PRG-1 critically depends on mitochondrial tethering. We did not detect PRG-1 in any of our experiments, indicating that, if PRG-1 interacts with PETISCO/ PUCH, this interaction is too transient to be detected through immunoprecipitations, similar to the PUCH–PETISCO interaction.

PUCH defines a type of ribonuclease, consisting of three subunits, each with one SLFN-like domain. Building on our findings, it is noteworthy that several mammalian proteins possess SLFN-folds. The *Slfn* gene cluster in mice has been described as an immunity locus, displaying high rates of sequence divergence[43]. Notably, a parental incompatibility syndrome, dysdiadochokinesia syndrome, has been linked to specific haplotypes of the *Slfn* gene cluster[43]. Given that the enzymatic activity of

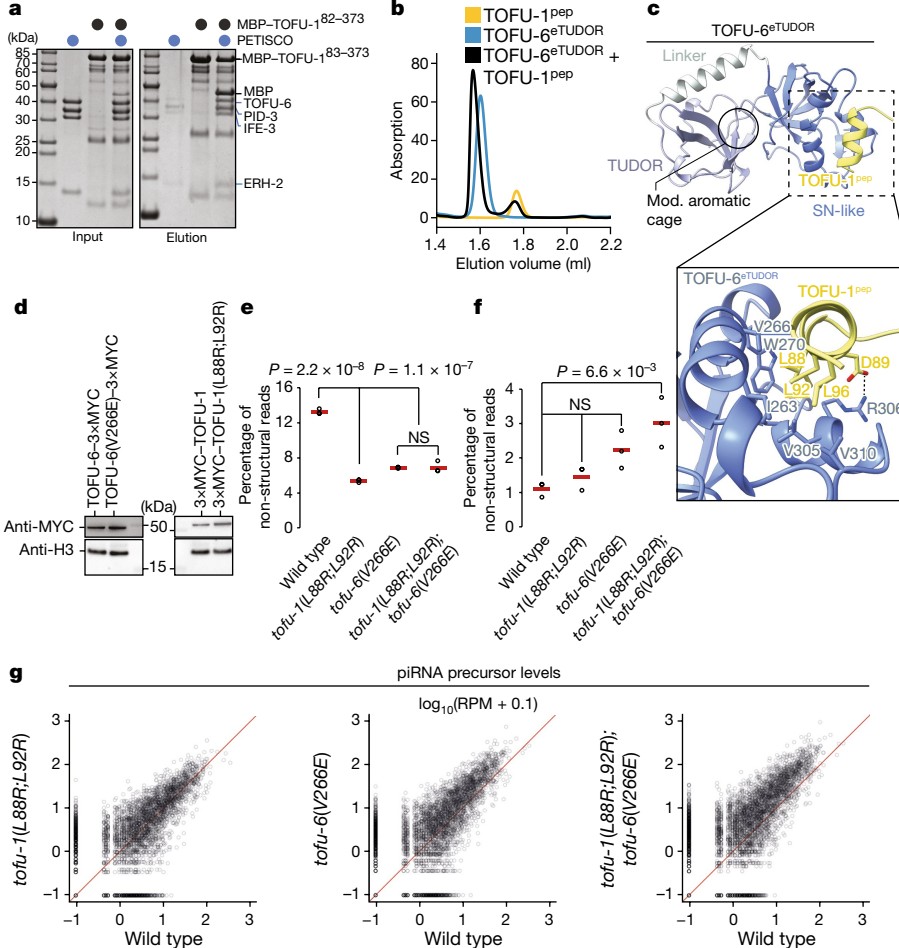

**Fig. 5 | TOFU-6 from PETISCO interacts with PUCH through TOFU-1.**
**a**, The interaction between TOFU-1 and PETISCO was analysed using amylose pull-down assays. Purified MBP-tagged TOFU-1[82–373] was incubated with excess PETISCO. Input and elution fractions were analysed by SDS−PAGE followed by Coomassie staining. **b**, Purified TOFU-6[eTUDOR], TOFU-1[pep] and a mixture thereof were analysed using size-exclusion chromatography. Chromatograms, TOFU-6[eTUDOR] (blue), TOFU-1[pep] (yellow) and TOFU-6[eTUDOR] + TOFU-1[pep] (black). Results from **a** and **b** were obtained in duplicates. **c**, The crystal structure of the TOFU-6[eTUDOR]−TOFU-1[pep] complex shown as a cartoon. The TOFU-6[eTUDOR] domain is shown in different shades of blue and TOFU-1[pep] in yellow. The magnified view shows the interaction interface; involved residues are shown as sticks. **d**, Western blot analysis of the expression levels of TOFU-1 and TOFU-6 for the indicated genotypes using anti-MYC and anti-H3 antibodies, followed by visualization using horseradish-peroxidase-linked secondary antibodies. The numbers indicate the approximate molecular mass (kDa). One out of two experiments is shown. **e**,**f**, Total mature (**e**) and precursor (**f**) type 1 piRNA levels in young adult hermaphrodites of indicated genotypes. $n = 3$ biological replicates. The red lines show the group means. $P$ values were calculated using one-way ANOVA followed by Tukey's HSD test. Note that mature and precursor reads derive from different libraries and their levels cannot be directly compared. NS, non-significant. **g**, The relative abundance of precursors from individual type 1 piRNA loci in young adult hermaphrodites of the indicated genotypes. $n = 3$ biological replicates. The underlying data are the same as in **f**. Raw data are provided in Supplementary Fig. 1.

PUCH requires association of three different SLFN-domain-containing subunits, one can hypothesize that, in mice, complexes between distinct paternal and maternal SLFN proteins may form active enzymes, of which the activity, or lack thereof, may trigger embryonic lethality. Another study in mice showed that a transposon-encoded non-coding RNA inhibits *Slfn* gene expression and therefore prevents overactivity of the innate immune system in response to virus infection[44]. Moreover, links between immunity and SLFN proteins are known in humans. For example, SLFN11 restrains translation of viral proteins during HIV infection by cleaving specific tRNAs[45]. Notably, SLFN11 is a protein with multiple activities. SLFN11 binds to single-stranded DNA, and it has been shown to also interfere with the replication of certain DNA viruses and to be recruited to stalled replication forks[38]. Furthermore, members of the *Orthopoxvirus* family, such as the monkeypox virus, contain a virulence factor that carries a single SLFN domain[46,47]. Even though the relevance of this specific domain for virulence has not been assessed, a role in host–pathogen interaction control seems likely. Mammalian proteins containing a single annotated SLFN domain can also be found[5], for example, SLFNL1 in human and mouse. This gene is testis-enriched and produces a protein with a single C-terminal SLFN fold. Its function is unclear, but it is dispensable for spermatogenesis[48]. Finally, a SLFN-related fold, the Smr domain, has been shown to act as a nuclease in RNA quality-control mechanisms, and this function can be traced back to the last universal common ancestor[49].

Overall, these activities, including the role that we identify in piRNA biogenesis, point to a deeply conserved role for SLFN-like domains in immunity- and stress-related mechanisms. Our results show that SLFN domains can form multimeric complexes and that multimerization can unveil highly specific nucleolytic activities. It is conceivable that combinations of proteins with SLFN-related folds may generate highly specific enzymes that help organisms to fight off infectious nucleic acids.

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

## Methods

### Worm culture

*C. elegans* strains were cultured on OP50 plates according to standard laboratory conditions[50]. For IP–MS experiments, worms were grown on high-density egg OP50 plates[51] and transferred to the standard OP50 plates for the last generation. The Bristol N2 strain was used as a reference wild-type strain. A list of the strains used is provided in Supplementary Table 3. Many aspects of this work made use of WormBase[52]. Blinding or randomization of strains and samples was not applied in this work. In all of the experiments, young adult hermaphrodite animals were used. Sample size calculations were not performed or required.

### CRISPR–CAS9-mediated genome editing

All protospacers were designed using CRISPOR (http://crispor.tefor.net) and afterwards confirmed using the Integrated DNA Technologies CRISPR–Cas9 guide RNA design checker. Protospacers were cloned into pRK2412 by site-directed ligase-independent mutagenesis. The Bristol N2 strain was used for microinjections unless stated otherwise. ssDNA oligonucleotides (IDT) were used as a repair template. Each of the repair templates has 35-nucleotide-long homology arms. The injection mix contained 50 ng μl⁻¹ plasmid encoding guide RNA for the gene of interest; 50 ng μl⁻¹ of plasmid containing Cas9 and *dpy-10*(cn64) or *unc-58*(e665) co-conversion guide RNA[53]; 750 nM of ssDNA oligonucleotide (repair template for gene of interest) and 750 nM of co-conversion ssDNA oligonucleotide. The strains RFK1692 and RFK1693 were obtained by injecting recombinant Cas9 protein (in house) and guide RNA molecule (IDT) as described previously[54]. A list of the protospacers and repair templates used is provided in Supplementary Table 3.

### Crosses with piRNA sensor

RFK1059 (*tofu-2(E216A)*) and RFK1481(*slfl-3(xf248)*) mutant hermaphrodite worms were crossed with males of the RFK1246 strain, which carries a *mut-7* deletion as well as the piRNA sensor[22]. Worms carrying piRNA sensor and *tofu-2(E216A)* or *slfl-3(xf248)* mutation and wild type for *mut-7* were selected by genotyping. A list of the genotyping primers is provided in Supplementary Table 3.

### Microscopy

Images of piRNA-sensor-carrying strains were obtained using the Leica DM6000B system. Young adults and adult worms were washed in a drop of M9 (22 mM KH₂PO₄, 42 mM Na₂HPO₄, 85 mM NaCl, 1 mM MgSO₄) and immobilized with 30 mM sodium azide in M9 buffer. Imaging of Bm4 cells was performed using the Leica TCS SP5 system with the LAS AF 2.7.3.9723 software. Images were processed using ImageJ and Adobe Illustrator.

### MS analysis

**Worm pellet preparation.** All IP–MS experiments were performed in quadruplicates. Worms, grown on the OP50 plates, were bleached (2% NaClO, 666 mM NaOH) into high-density egg plates, grown until the gravid adult stage and bleached again. The embryos were left to hatch in M9 buffer (22 mM KH₂PO₄, 42 mM Na₂HPO₄, 85 mM NaCl, 1 mM MgSO₄), L1-stage worms were seeded on standard OP50 plates and collected at the young adult stage. The worms were washed three times with M9 buffer and one time with cold sterile water. Worm aliquots (200 μl) were pelleted and frozen in liquid nitrogen and stored at −80 °C.

**Lysis preparation.** A total of 200 μl of synchronized young adult worms was thawed on ice and resuspended in 250 μl of 2× lysis buffer (50 mM Tris HCl pH 7.5, 300 mM NaCl, 3 mM MgCl₂, 2 mM DTT, 2 mM Triton X-100, 2× cOmplete Mini, EDTA-free (Roche, 11836170001)) and 50 μl of sterile water. The Bioruptor Plus (Diagenode) sonicator was used to lyse the worms (10 cycles 30/30 s, high energy, 4 °C). After pelleting, the supernatant was accurately removed without the lipid phase. Finally, the protein concentration of the lysate was determined using the Pierce BCA Protein Assay Kit (Thermo Fisher Scientific, 23225).

**Immunoprecipitation.** For anti-HA immunoprecipitations, 550 μl of worm lysate containing 0.75 mg protein was resuspended in a final volume of 550 μl of 1× lysis buffer. Anti-HA immunoprecipitation was performed using 2 μg of custom-made anti-HA antibodies (mouse, 12CA5). The lysate was incubated with the antibodies for 2 h at 4 °C. For each sample, 30 μl of protein G magnetic beads (Dynabeads, Invitrogen) was washed three times in washing buffer (25 mM Tris HCl pH 7.5, 150 mM NaCl, 1.5 mM MgCl₂, 1 mM DTT, 1 mM Triton X-100, cOmplete Mini, EDTA-free (Roche, 11836170001)). Subsequently, equilibrated beads were added to the lysis and incubated for an additional hour at 4 °C by end-over-end rotation. Finally, beads were washed 6 times with wash buffer, resuspended in 2× NuPAGE LDS sample buffer (containing 200 mM DTT) and boiled for 15 min at 95 °C.

To identify TOFU-2–HA and TOFU-2(E216A)–HA, the samples were separated on a 4–12% NOVEX NuPAGE gradient SDS gel (Thermo Fisher Scientific) for 10 min at 180 V in 1× MES buffer (Thermo Fisher Scientific). Proteins were fixed and stained with Coomassie G250 Brilliant Blue (Carl Roth). The gel lanes were cut, minced into pieces and transferred to an Eppendorf tube. Gel pieces were destained with a 50% ethanol/50 mM ammonium bicarbonate (ABC) solution. Proteins were reduced in 10 mM DTT (Sigma-Aldrich) for 1 h at 56 °C and then alkylated with 5 mM iodoacetamide (Sigma-Aldrich) for 45 min at room temperature. Proteins were digested with trypsin (Sigma-Aldrich) overnight at 37 °C. Peptides were extracted from the gel by two incubations with 30% ABC/acetonitrile and three subsequent incubations with pure acetonitrile. The acetonitrile was subsequently evaporated in a concentrator (Eppendorf) and loaded onto StageTips[55] for desalting and storage.

For MS analysis, peptides were separated on a 20 cm self-packed column with a 75 μm inner diameter filled with ReproSil-Pur 120 C18-AQ (Dr. Maisch) mounted to an EASY HPLC 1000 (Thermo Fisher Scientific) system and sprayed online into an Q Exactive Plus mass spectrometer (Thermo Fisher Scientific). We used a 94 min gradient from 2 to 40% acetonitrile in 0.1% formic acid at a flow rate of 225 nl min⁻¹. The mass spectrometer was operated with a top 10 MS/MS data-dependent acquisition scheme per MS full scan. MS raw data were searched using the Andromeda search engine[56] integrated into the MaxQuant suite (v.1.6.5.0)[57] using the UniProt *C. elegans* database (August 2014; 27,814 entries). In both analyses, carbamidomethylation at cysteine was set as a fixed modification, whereas methionine oxidation and protein *N*-acetylation were considered variable modifications. The match-between-run option was activated. Before bioinformatic analysis, reverse hits, proteins only identified by site, protein groups based on one unique peptide and known contaminants were removed.

For the further bioinformatic analysis, the label-free quantification values were log₂-transformed and the median across the replicates was calculated. This enrichment was plotted against the −log₁₀-transformed *P* value (Welch's *t*-test) using the ggplot2 package in the R environment.

### RNA isolation and small RNA sequencing

Worms were grown at 20 °C, synchronized by bleaching (2% NaClO, 666 mM NaOH) and were left to hatch overnight in M9 buffer. Next, L1-stage worms were seeded onto OP50 plates and collected as young adults. For RNA extraction, 500 μl of TRIzol LS (Thermo Fisher Scientific, 10296-028) was added to the 50 μl worm aliquot, and five cycles of freezing in liquid nitrogen and thawing in a 37 °C water bath were performed. The samples were centrifuged for 5 min at 21,000g at room temperature, and the supernatant was collected. An equal volume of 100% ethanol was added to the supernatant before proceeding with RNA extraction using the Direct-zol RNA MicroPrep (Zymo Research) kit. RNA was eluted into 13 μl of nuclease-free water (Ambion Invitrogen)

and each sample was divided into two aliquots for piRNA-precursor and mature piRNA library preparation.

**CIP/RppH treatment for piRNA precursors.** CIP treatment of 1.5 µg of isolated RNA was performed in rCutSmart Buffer (B6004S) using 3 µl of Quick CIP (M0525L) in a 40 µl reaction. The reaction was incubated at 37 °C for 20 min, followed by heat-inactivation for 2 min at 80 °C. The CIP-treated RNA was subjected to another round of purification using the Direct-zol RNA MicroPrep (Zymo Research) kit. RppH (NEB) treatment was performed with a starting amount of 500 ng.

**Library preparation and sequencing.** Next-generation sequencing library preparation was performed using the NEXTflex Small RNA-Seq Kit V3 following step A to step G of Bioo Scientific's standard protocol. Amplified libraries were purified by running an 8% TBE gel and size-selected for 15–40 nucleotides. Libraries were profiled using a High Sensitivity DNA Chip on the 2100 Bioanalyzer (Agilent Technologies), quantified using the Qubit dsDNA HS Assay Kit in the Qubit 2.0 Fluorometer (Life Technologies) and sequenced on the Illumina NextSeq 500/550 system.

**Next-generation sequencing data analysis.** The raw sequence reads in FastQ format were cleaned of adapter sequences and size-selected for 18–35-nucleotide inserts (plus 8 random adapter bases) using Cutadapt v.4.0 (http://cutadapt.readthedocs.org) with the parameters '-a TGGAATTCTCGGGTGCCAAGG -m 26 -M 43'. Data quality was assessed with FastQC v.0.11.9 (https://github.com/s-andrews/FastQC) and MultiQC v.1.9 (https://multiqc.info/). Read alignment to the *C. elegans* genome (Ensembl WBcel235/ce11 assembly) allowing for one mismatch and reporting one best alignment for each read while concomitantly removing the 2×4 nucleotide random adapter bases was performed using Bowtie v.1.3.1 (http://bowtie-bio.sourceforge.net) with the parameters '-v 1 -M 1 -y --best --strata --trim5 4 --trim3 4 -S' and the SAM alignment files were converted into sorted BAM files using Samtools v.1.10 (http://www.htslib.org). *C. elegans* WBcel235/ce11 gene annotation in GTF format was downloaded from Ensembl release 96 (https://ftp.ensembl.org/pub/release-96/gtf/caenorhabditis_elegans/). Only type 1 piRNAs (21ur loci) were annotated in the GTF file (gene_biotype "piRNA"). The annotation of type 2 piRNAs was published[24] and their genome coordinates were lifted from the ce10 to ce11 assembly using LiftOver (http://genome-euro.ucsc.edu/cgi-bin/hgLiftOver). Aligned reads were assigned to small RNA loci and classes using Samtools v.1.10, GNU Awk v.5.1.0 and Subread featureCounts v.2.0.0 (https://subread.sourceforge.net/). Structural reads aligned in sense orientation to rRNA, tRNA, snRNA and snoRNA loci were excluded from further analysis. Mature piRNAs were stringently defined as reads of 21 nucleotides in length starting with T and fully overlapping with annotated piRNA genes in sense orientation. To accomplish this selection, first all of the aligned 21 nucleotide reads starting with T were isolated using 'awk '$6 ~ /21 M/' | awk '$2 = = 0' | awk '$10 ~ /^T/'' for forward-strand reads and 'awk '$6 ~ /21 M/' | awk '$2 = =16' | awk '$10 ~ /A$/'' for reverse-strand reads. The combined SAM files were then converted into BAM format using Samtools and used for stringent counting of type 1 or type 2 piRNAs using 'featureCounts -s 1 -M --minOverlap 21'. As piRNA gene annotation corresponds to mature piRNA sequences, piRNA precursors were stringently defined as reads of 23–35 nucleotides in length starting 2 nucleotides upstream of the annotated 5′ ends of mature piRNAs in sense orientation. This was achieved using 'featureCounts -s 1 -M --read2pos 5' and a GTF file with all genomic positions 2 nucleotides upstream of 21ur loci (type 1 mature piRNAs). An alternative (relaxed) assignment of mature and precursor piRNAs was also tested by counting all 18–35 nucleotide reads overlapping in sense with 21ur piRNA loci—the resulting quantification patterns were similar and all conclusions remained unchanged. We prefer the stringent definition approach to avoid misassignment of residual mature 21Us as piRNA precursors in the precursor

libraries and vice versa in the mature libraries. For maximal specificity, a small number (3.6%) of ambiguous 21ur piRNA loci colocalizing on the same strand with miRNAs, snoRNAs or other RNA exons was excluded from analysis. The relative abundance of mature and precursor piRNAs was normalized to the number of non-structural 18–35 nucleotide reads in each sample. Coverage tracks of aligned 18–35 nucleotide reads overlapping in sense with piRNA genes were produced using Bedtools genomeCoverageBed v.2.27.1 (http://bedtools.readthedocs.io) and kentUtils bedGraphToBigWig v.385 (https://github.com/ucscGenomeBrowser/kent). The tracks were normalized on the basis of all non-structural reads in each sample and visualized on the IGV genome browser v.2.15.4 (https://igv.org/).

### 3′ RNA radioactive labelling
3′-end labelling of substrate RNA (the sequence is shown in Supplementary Table 3) was performed in a 25 µl reaction containing 2.5 µl DMSO, 2.5 µl of T4 ligase buffer (NEB), 1 µl of T4 ligase (NEB), 2.5 µl 10 mM ATP (NEB), 1 µl of synthetic RNA precursor (5 pmol µl$^{-1}$). The reaction was mixed and 2.5 µl of [5′-$^{32}$P]pCp (SCP-111, Hartmann analytic) was added before overnight incubation at 16 °C. Finally, the labelled RNA was purified using G25 columns (Cytiva) according to the manufacturer's protocol. The 3′-end-labelled synthetic RNA precursor was used for in vitro cleavage assays and in EMSAs.

### 5′ RNA radioactive labelling
A total of 5 pmol synthetic RNA oligonucleotide was labelled with ATP, [γ-$^{32}$P] (PerkinElmer) using T4 PNK (NEB), according to the manufacturer's protocol. The sequences of the RNA substrates are provided in Supplementary Table 3.

### Plasmids
Full-length CeTOFU-2 was amplified from N2 cDNA and was inserted by restriction-based cloning into the pBEMBL vector (gift from R. Pillai) in which expression of an N-terminal eGFP tag is driven by the OpIE2 promoter. Likewise, CeTOFU-1 was inserted into a vector containing a C-terminal 3×Flag-mCherry cassette. CeSLFL3 and CeSLFL4 were inserted into a vector backbone containing an N-terminal HA tag. All of the primers, vector backbones and detailed cloning strategies are provided in the Supplementary Information.

### BmN4 cell culture and transfection
BmN4 cells were cultured at 27 °C in IPL-41 insect medium (Gibco) supplemented with 10% FBS and 0.5% penicillin–streptomycin (Gibco). Then, 24 h before transfection, around $4 × 10^6$ cells were seeded into a 10 cm dish (using one 10 cm dish for each condition in the cleavage reaction). Cells were transfected with 10 µg of each plasmid DNA using XtremeGene HP (Roche) transfection reagent, according to the manufacturer's instructions. Then, 72 h after transfection, cells were collected, washed twice in ice-cold PBS and pelleted by centrifugation at 500g for 5 min at 4 °C. The cell line was obtained from R. Pillai in 2015. BmN4 cells were obtained from T. Kusakabe. Further details are available online (https://www.cellosaurus.org/CVCL_Z634). It was not authenticated and was not tested for mycoplasma.

### GFP immunoprecipitation from BmN4 cells
Approximately $4 × 10^6$ BmN4 cells were collected from each 10 cm dish (see above), washed once in 5 ml ice-cold PBS and once more in 1 ml ice-cold PBS. Subsequently, cells were pelleted by centrifugation for 5 min at 500g at 4 °C and frozen at −80 °C. Directly before use, BmN4 cell pellets were thawed on ice and lysed in 1 ml IP-150 lysis buffer (30 mM HEPES (pH 7.4), 150 mM KOAc, 2 mM Mg(OAc)$_2$ and 0.1% IGEPAL freshly supplemented with EDTA-free protease inhibitor cocktail and 5 mM DTT) for 1 h by end-over-end rotation at 4 °C. Cells were further lysed by passing the lysate ten times through a 20-gauge syringe needle followed by five passes through a 30-gauge needle. Cell debris was pelleted by

centrifugation at 17,000*g* for 20 min at 4 °C. The supernatant fractions were collected and processed for GFP immunoprecipitation using GFP-Trap beads (Chromotek). The GFP-Trap beads (15 µl bead suspension per reaction) were washed three times in 1 ml of IP-150 lysis buffer. Equilibrated beads were subsequently incubated with the BmN4 cell lysate and incubated overnight by end-over-end rotation at 4 °C. The next day, immunoprecipitated complexes were washed five times using 1 ml of IP-150 lysis buffer and were subsequently used for in vitro cleavage assays or for immunodetection using western blot analysis.

### Western blotting

**BmN4 cells.** Samples were prepared in 1× Novex NuPage LDS sample buffer (Invitrogen) supplemented with 100 mM DTT and were heated at 95 °C for 10 min before resolving on a 4–12% Bis-Tris NuPage NOVEX gradient gel (Invitrogen) in 1× Novex NuPAGE MOPS SDS running buffer (Invitrogen) at 140 V. Separated proteins were transferred to a nitrocellulose membrane (Amersham) overnight at 20 V using 1× NuPAGE transfer buffer (Invitrogen) supplemented with 10% methanol. The next day, the membrane was incubated for 1 h in 1× PBS-Tween-20 (0.05%) supplemented with 5% skimmed milk and incubated for 1 h with primary antibodies diluted in PBS-Tween-20 (1:1,000 monoclonal anti-Flag M2, F3165, Sigma-Aldrich; 1:1,000 monoclonal anti-GFP antibodies (B-2), Santa Cruz, sc-9996, K1115; 1:1,000 monoclonal anti-HA (12CA5, in house); 1:1,000 anti-actin (A5060) rabbit monoclonal antibodies, Sigma-Aldrich). Subsequently, the membrane was washed three times for 5 min in PBS-Tween-20 before incubation with secondary antibodies, using 1:10,000 IRDye 800CW goat anti-mouse and IRDye 680LT donkey anti-rabbit IgG (LI-COR) and imaged on the Odyssey CLx imaging system (LI-COR). The blots were scanned using Image Lab (v.6.0.1).

**Worm lysates.** Strains RFK 1269 and RFK1280 were grown and lysed as described in the 'MS analysis' section. A total of 15 µg of protein was mixed with 2× gel loading buffer (2× Novex NuPage LDS sample buffer (Invitrogen), supplemented with 200 mM DTT) and heated at 95 °C for 10 min before resolving on a 4–12% Bis-Tris NuPage NOVEX gradient gel (Invitrogen) in 1× Novex NuPAGE MOPS SDS Running Buffer (Invitrogen) at 150 V. Separated proteins were transferred to nitrocellulose membrane (Amersham) 1 h at 120 V using 1× NuPAGE transfer buffer (Invitrogen) supplemented with 10% methanol. The membrane was incubated for 30 min in 1× PBS-Tween-20 (0.05%) supplemented with 5% skimmed milk, cleaved and incubated overnight with primary antibodies diluted in PBS-Tween-20 (1:1,000 monoclonal anti-HA (12CA5, in house); 1:1,000 anti-H3 (H0164, Sigma-Aldrich) rabbit polyclonal antibodies). Subsequently, the membrane was washed five times for 5 min in PBS-Tween-20 before incubation with secondary antibodies, using 1:10,000 horse anti-mouse HRP-linked antibody (7076, Cell Signaling) and goat anti-rabbit HRP-linked antibodies (7074, Cell Signaling) and imaged using the SuperSignal West Pico Plus (Thermo Fisher Scientific) kit.

For strains RFK1057, RFK1506, RFK1692 and RFK1693, 50 young adult worms were picked into 13 µl of M9 buffer, 5 µl of 4× Novex NuPage LDS sample buffer (Invitrogen) and 2 µl of 1 M DTT, boiled for 30 min at 95 °C and loaded onto the 4–12% Bis-Tris NuPage NOVEX gradient gel (Invitrogen). Gel run, transfer, staining and imaging were performed as described above; anti-MYC (1:1,000, mouse anti-MYC (9B11), 2276S, Cell Signaling) antibodies were used. Blots were scanned using Image Lab (v.6.0.1).

### In vitro cleavage assay

The PUCH complex used for in vitro cleavage assays was obtained using two different methods. The full-length PUCH complex was obtained from GFP immunoprecipitates using BmN4 cell lysates (see above), whereas the minimal catalytic complex (mini-PUCH) was purified from *E. coli*.

For the in vitro cleavage assays performed with immunoprecipitated material from BmN4 cells, beads were washed in the cleavage buffer

(CB) containing 40 mM Tris-HCl, pH 8.0, 20 mM KCl, 11 mM MgCl₂ and 2 mM DTT. The beads were subsequently resuspended in 10 µl of CB and incubated with 0.2 pmol of the labelled RNA substrate for 1 h at room temperature.

For cleavage assays with mini-PUCH purified from *E. coli*, 0.2 pmol of labelled RNA substrate was incubated in 10 µl CB with 27 nM mini-PUCH protein complex (final concentration) at 20 °C for 30 min.

The cleavage reaction was terminated by adding 1 µl of 20 mg ml⁻¹ proteinase K. One volume of the 2× RNA gel loading dye (Thermo Fisher Scientific, R0641) was added and the RNA was resolved on a 15% TBE-UREA gel (Novex) for 90 min at 180 V with 1× TBE as the running buffer.

**Substrate specificity test of PUCH complex.** Capped RNA oligonucleotides were labelled at the 3′ end and 0.2 pmol (1 µl) of RNA per sample was used in the cleavage reaction. For the reaction with immunoprecipitated material, to obtain 5′P-containing piRNA precursor oligonucleotide, 5′OH-piRNA precursor had been labelled at the 3′ end as described above. After labelling, 5′P was created by T4 PNK treatment (NEB, M0201S) according to the NEB T4 PNK protocol. For the reaction with mini-PUCH 5′OH-piRNA precursor, the oligonucleotide had been labelled at the 5′ end as described above.

**CAU and AAU substrate comparison.** A total of 18 µl of 0.2 pmol µl⁻¹ RNA substrate (AAU or CAU) was added to the 162 µl of CB, containing recombinant mini-PUCH at a final concentration 27 nM. The samples were transferred to 20 °C and the samples for each timepoint were taken. The reaction was stopped by adding proteinase K. Images were processed using ImageJ.

**Competition assay with cold RNA.** A total of 10 µl of CB with 27 nM mini-PUCH was added to a 2 µl mix of 0.2 pmol labelled AAU substrate and 0.4 pmol cold RNA of choice. Cleavage reactions were incubated at 20 °C for 15 min and were stopped by adding protease K.

**Analysis of divalent cations as a cofactor of the PUCH complex.** To test metal requirements, BmN4 cells were lysed in EDTA + lysis buffer (30 mM HEPES (pH 7.4), 150 mM KOAc, 1 mM EDTA and 0.1% IGEPAL freshly supplemented with EDTA-free protease inhibitor cocktail and 5 mM DTT) after the immunoprecipitation, beads were washed five times with EDTA + lysis buffer followed by one wash in CB containing 1 mM EDTA. Next, the cleavage reactions were performed in CB containing MgCl₂, ZnCl₂, MnCl₂ or CaCl₂ at the indicated concentrations (1, 4 or 11 mM), or with no divalent metals at all.

**Ligation of RNA oligo to the PUCH cleavage product.** A total of 2 pmol of labelled RNA was incubated in 35 µl of CB containing mini-PUCH (or mutated mini-PUCH) at a final concentration of 40 nM and was incubated at 20 °C for 1 h. Afterwards, 3 volumes of TRIzol LS reagent (Thermo Fisher Scientific, 10296-028) was added, and RNA was purified using Direct-zol RNA MicroPrep (Zymo Research) according to the manufacturer's protocol. Next, the RNA was ligated to 10 pmol of 5′OH-rGrUrCrUrGrUrUrArA-OH3′ oligonucleotide using T4 RNA ligase according to the manufacturer's protocol. After 16 h of incubation at 16 °C, the reaction was terminated by proteinase K and RNA was resolved on a 15% TBE-UREA gel (Novex) for 90 min at 180 V with 1× TBE as the running buffer.

**PUCH complex cleavage activity in the presence of PETISCO.** A total of 16 µl of 3′-end-labelled piRNA precursor (0.2 pmol µl⁻¹) was incubated with five times molar excess of PETISCO protein complex on ice for 1 h in 160 µl of CB. After the incubation, PUCH-immunoprecipitate-containing beads were added, and the sample was split into two tubes. The same procedure was performed in parallel for RNA incubated without PETISCO. Reactions were incubated at 20 °C with mild shaking, and

10 μl samples were taken for each timepoint. The same experiment was performed with recombinant mini-PUCH at the concentrations described for cleavage reactions.

Gels were scanned using the Typhoon FLA 9500 system (software version V.0 build 1.0.0.185).

### EMSA

A total of 0.2 pmol of capped piRNA precursor, 5′P piRNA precursor and 5′OH-piRNA precursor was incubated with recombinant proteins of PETISCO complex, containing IFE-3, TOFU-6, ERH-2 and PID-3 (ref. 28) in a concentration range from 75 pM to 1.44 μM, in 10 μl of binding buffer (20 mM HEPES pH 7.5, 150 mM NaCl) for 1 h at the room temperature. After the incubation, each sample was mixed with 15% Ficoll with bromophenol blue. Native 6% TBE gel was pre-run for 30 min at 180 V at room temperature in 1× TBE, and the samples were resolved for 2 h. Gels were scanned using the Typhoon FLA 9500 system (software version V.0 build 1.0.0.185).

### Recombinant protein production in *E. coli*

PETISCO and its subunits (IFE-3, TOFU-6, PID-3, ERH-2) were purified and reconstituted as described previously[28]. Using ligation-independent cloning, genes encoding TOFU-1, TOFU-2 and SLFL-3 were cloned into modified pET vectors. All proteins were produced as an N-terminal His-Tagged fusion protein with varying fusion tags that can be removed by the addition of 3C protease. Proteins or protein complexes were produced in the *E. coli* BL21(DE3) derivate strains in terrific broth medium. In brief, cells were grown at 37 °C, and when the culture reached an optical density at 600 nm ($OD_{600}$) of 2–3, the temperature was reduced to 18 °C. After 2 h at 18 °C, 0.2 mM IPTG was added to induce protein production for 12–16 h overnight.

### Co-expression pull-down assays

For interaction studies using the co-expression co-purification strategy, two plasmids containing the gene of interest and different antibiotic-resistance markers were co-transformed into BL21(DE3) derivative strains to allow co-expression. A total of 50 ml of cells was grown in TB medium under shaking at 37 °C and, when the culture reached an $OD_{600}$ of 2–3, the temperature was reduced to 18 °C. Protein production was induced after 2 h at 18 °C through the addition of 0.2 mM IPTG for 12–16 h overnight. Cells were collected by centrifugation and the cell pellets were resuspended in 2 ml of lysis buffer (50 mM sodium phosphate, 20 mM Tris/HCl, 250 mM NaCl, 10 mM imidazole, 10% (v/v) glycerol, 0.05% (v/v) NP-40, 5 mM 2-mercaptoethanol pH 8.0) per gram of wet cell mass. Cells were lysed by ultrasonic disintegration, and insoluble material was removed by centrifugation at 21,000*g* for 10 min at 4 °C. For Strep-Tactin pull-downs, 500 μl of supernatant was applied to 20 Strep-Tactin XT resin (IBA Lifesciences); for MBP pull-downs, 500 μl supernatant was applied to 20 μl amylose resin (New England Biolabs) and incubated for 2 h at 4 °C. Subsequently, the resin was washed three times with 500 μl of lysis buffer. The proteins were eluted in 50 μl of lysis buffer supplemented with 10 mM maltose or 50 mM biotin in the case of amylose beads or Strep-Tactin XT beads, respectively. Input material and eluates were analysed by SDS–PAGE and Coomassie brilliant blue staining.

### Pull-down assays with purified proteins

To analyse protein interactions with purified proteins, appropriate protein mixtures (bait 10–20 μM, prey in 1.2-fold molar excess) were incubated in binding buffer containing 20 mM Tris/HCl (pH 7.5), 150 mM NaCl, 10% (v/v) glycerol, 0.05% (v/v) NP40, 1 mM DTT for 30 min at 4 °C. Subsequently, the indicated beads were added to the protein mixtures were then incubated with the indicated beads for 2 h on ice: Glutathione Sepharose beads (CubeBiotech), Amylose Sepharose beads (New England Biolabs) and Strep-Tactin XT beads (IBA). Subsequently, the beads were washed three times with 200 μl binding buffer, and the retained

material was eluted with 0.05 ml incubation buffer supplemented with 20 mM of reduced glutathione, 10 mM maltose or 50 mM biotin. Input material and eluates were analysed using SDS–PAGE and Coomassie brilliant blue staining.

### Purification of the trimeric mini-PUCH

To reconstitute minimal PUCH, TOFU-1 (residues 160–373), TOFU-2 (residues 200–433) and SLFL-3 (residues 1–345) were co-expressed in BL21(DE3) cells. In the case of inactive minimal PUCH, an inactive TOFU-2 mutant (residues 200–433, E216A) was used. TOFU-1 carried an N-terminal His10-MBP tag, TOFU-2 an N-terminal His10-MBP and a C-terminal Strep II tag and SLFL-3 an N-terminal His6-GST tag. Cells were grown at 37 °C and, when the culture reached an $OD_{600}$ of 2–3, the temperature was reduced to 18 °C. After 2 h at 18 °C, 0.2 mM IPTG was added to induce protein production for 12–16 h overnight. All of the purification steps were performed on ice or at 4 °C. Cells were lysed by sonication in lysis buffer (50 mM sodium phosphate, 20 mM Tris/HCl, 500 mM NaCl, 20 mM imidazole, 10% (v/v) glycerol and 5 mM 2-mercaptoethanol at pH 8.0). PUCH was purified by immobilized metal affinity chromatography (IMAC) using a 5 ml $Ni^{2+}$-chelating HisTrap FF column (Cytiva). Proteins were eluted with lysis buffer supplemented with 500 mM imidazole and dialysed overnight against 20 mM Tris/HCl, 150 mM NaCl, 10% (v/v) glycerol and 5 mM 2-mercaptoethanol at pH 7.5. After dialysis, PUCH was subjected to heparin affinity chromatography on a 5 ml HiTrap Heparin HP (Cytiva) followed by size-exclusion chromatography on the HiLoad Superdex 200 16/600 (Cytiva) column in 20 mM Tris/HCl pH 7.5, 150 mM NaCl, 10% (v/v) glycerol and 2 mM DTT.

### Differential scanning fluorimetry

The thermal stability of mini-PUCH WT and the E216 mutant versions was determined using differential scanning fluorimetry. In a total volume of 25 μl, 0.1 mg ml$^{-1}$ mini-PUCH was mixed with a final concentration 5× SYPRO Orange (Thermo Fisher Scientific) in a buffer containing 20 mM Tris/HCl pH 7.5, 150 mM NaCl, 10% (v/v) glycerol and 2 mM DTT. Unfolding transitions were measured using the CFX96 Touch real-time PCR machine (Bio-Rad) by increasing the temperature from 15 °C to 95 °C in 0.5 °C increments (10 s hold time). Fluorescence was measured every 0.5 °C. Data analysis was performed using the CFX Manager software (Bio-Rad) included with the real-time PCR machine.

### Analytical size-exclusion chromatography

Purified proteins were incubated alone or in different mixtures at concentrations between 20 μM and 40 μM (total volume of 50 μl) in size-exclusion buffer (20 mM Tris/HCl pH 7.5, 150 mM NaCl, 2 mM DTT) as indicated in the figure legends. The samples were incubated for 1 h on ice to allow complex formation. Complex formation was assayed by comparing the elution volumes in size-exclusion chromatography on the Superdex 200 Increase 3.2/300 (Cytiva) column. The size-exclusion chromatography peak fractions were analysed using SDS–PAGE and visualized by Coomassie brilliant blue staining. Unicorn7 software was used for data acquisition, and Datagraph5 was used for plotting.

### ITC analysis

Isothermal titration calorimetry (ITC) experiments to quantitatively analyse the interaction between TOFU-1 peptide (residues 82–113) and the TOFU-6 eTUDOR domain (residues 119–314) interaction were performed using the PEAQ-ITC Isothermal titration calorimeter (Malvern). The TOFU-1$^{82–113}$ peptide does not contain tyrosine or tryptophane residues. To be able to determine the concentration precisely, we engineered a TOFU-1 peptide (TOFU-1$^{W-82–113}$) that contains a Tryptophan residue at the N terminus. Data processing and analysis was performed using the PEAQ-ITC software (Malvern). Before the measurements, the samples were dialysed overnight simultaneously against

1 l of ITC buffer (20 mM Tris, 250 mM NaCl, 0.5 mM TCEP, pH 7.50). TOFU-1$^{W-82-113}$ (the reactant) samples were concentrated to 45–48 µM and TOFU-6$^{eTUDOR}$ (the injectant) to 400–450 µM. Titrations were carried out at 25 °C with 2 µl of the injectant per injection added to 200 µl of reactant cell solution. The reported $K_d$ and stoichiometry are the average of three experiments, and the reported experimental error is the s.d. The MicroCal PEAQ-ITC Control Software v.1.41 was used for data acquisition.

## TOFU-6$^{eTUDOR}$ and TOFU-1$^{pep}$ crystallization

Purified TOFU-6$^{eTUDOR}$ and TOFU-1$^{W-82-113}$ were mixed with TOFU-1$^{W-82-113}$ being in 1.5-fold molar excess and subjected to size-exclusion chromatography on the HiLoad Superdex S75 16/600 (Cytiva) column equilibrated in 20 mM Tris/HCl, 150 mM NaCl, 2 mM DTT pH 7.5. The complex-containing fractions were concentrated to 10 mg ml$^{-1}$ by ultrafiltration. Crystallization trials were performed at 4 °C and 22 °C at 8–10 mg ml$^{-1}$ using a vapour-diffusion set-up. Drops were set up using the mosquito Crystallization Robot (SPT Labtech) on 96-Well 2-Drop MRC Crystallization Plates (Swissci) by mixing the protein complex and crystallization solution at 200 nl:200 nl and 400 nl:200 nl ratios.

Small crystals grew at 4 °C in various conditions of the Morpheus Screen[58]. Several rounds of microseed matrix screening yielded larger crystals. The best crystals grew in 0.2 M Na bromide, 0.1 M Bis Tris propane pH 7.5, 20% (w/v) PEG 3350 at 22 °C in the PACT screen[59]. Crystals were soaked with a mother liquor supplemented with 20% (v/v) glycerol for cryoprotection and then frozen in liquid nitrogen.

## Data collection, structure determination and refinement

Data were collected at the ESRF (Grenoble, France) beamline ID30A-3 on 26 September 2021 (https://doi.org/10.15151/ESRF-DC-1033968485).

Data were processed with autoPROC[60] using XDS[61] and AIMLESS[62]. Phases were determined by molecular replacement using the Alpha-Fold model of the *C. elegans* TOFU-6 eTUDOR domain (residues 120–314) (https://alphafold.ebi.ac.uk/entry/Q09293). Molecular replacement was performed using Phaser[63] within Phenix[64]. The model was processed using Phenix (process predicted model) to translate the pLDDT values to *B* factors and to remove flexible regions. After molecular replacement, the model was automatically built using Buccaneer[65], manually completed with COOT[66] and refined using phenix.refine[67]. The model quality was assessed using molprobity[68] and PDB-REDO[69]. The refined model has a clashscore of 1.17 and 98.56% of the residues fall into Ramachandran-favoured and 1.44% into Ramachandran-allowed regions. Data collection and refinement statistics are listed in Extended Data Table 1. Molecular graphics of the structures were prepared using UCSF ChimeraX[70]. Coordinates and structure factors have been deposited in the PDB under accession code 8BY5.

## Protein complex structure prediction

Initial structural homology was detected using HHPRED[71].

The prediction of protein complex structures was performed using AlphaFold[72–74] v.2.1.0 on the Colab notebook (ColabFold)[75] (https://colab.research.google.com/github/sokrypton/ColabFold/blob/main/AlphaFold2.ipynb).

The following settings were used: template_mode (none), msa_mode (MMSeq2 (UniRef+Environmental), pair_mode (unpaired + paired), model_type (AlphaFold2-multimer-v2). In the case of the tetrameric PUCH, 3 recycles were used; for the trimeric PUCH, 48 recycles were used. Protein sequences were obtained from UniProt and, for initial complex predictions, full-length sequences for all four proteins (TOFU-1, TOFU-2, SLFL-3 and SLFL-4) were used. The predicted models and the predicted alignment error score were visualized and analysed using ChimeraX[70]. Predicted complexes contained either SLFL-3 or SLFL-4. As SLFL-3 and SLFL-4 are paralogues that are 90% identical and 93%

similar at the protein-sequence level, we focused on predictions of the trimeric PUCH containing TOFU-1, TOFU-2 and SLFL-3. For the prediction of the core PUCH, the following residue boundaries were used: TOFU-1 residues 156–373, encompassing the SLFN domain with an N-terminal extension; TOFU-2 residues 200–433, encompassing the SLFN domain and two C-terminal alpha helices; SLFL-3 residues 103–300, encompassing the SLFN domain.

## Reporting summary

Further information on research design is available in the Nature Portfolio Reporting Summary linked to this article.

## Data availability

Sequencing data are available at the NCBI Sequence Read Archive under accession number PRJNA925182. The MS proteomics data have been deposited at the ProteomeXchange Consortium through the PRIDE[76] partner repository under dataset identifier PXD039502. Coordinates and structure factors of the TOFU-6$^{eTUDOR}$–TOFU-1$^{pep}$ complex structure have been deposited at the PDB (9G6Z). Wormbase WS289 was used in this study. UniProt was regularly used, and the most recent version was always used. The AlphaFold database (https://alphafold.ebi.ac.uk/) was used in this work. Source data are provided with this paper.

## Code availability

All computational tools are public and details on their use are provided in the Methods.

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

**Acknowledgements** We thank all of the members of the Ketting and Falk laboratories for discussions. This work was funded by the Deutsche Forschungsgemeinschaft (DFG, German Research Foundation) project IDs 439669440-TRR319, 252386272 and 504320275 (to R.F.K.) and the Austrian Science Fund (FWF) I6110-B (to S.F.). A.W.B. was supported by the Peter und Traudl Engelhorn Foundation. Some strains were provided by the *Caenorhabditis* Genetics Center (CGC), funded by NIH Office of Research Infrastructure Programs (P40 OD010440). We acknowledge support by the members of the IMB Genomics Core Facilities and the use of the NextSeq 500 system (funded by the DFG, INST P#329045328) and M. Möckel from the IMB Protein Production Core Facilities. We thank J. Schreier, K. Holleis and L. Miksch for their contribution to the early stages of this project; and the beamline scientists from the European Synchrotron Radiation Facility (ESRF) beamline ID30A-3 (Grenoble, France) for support with data collection.

**Author contributions** N.P. designed, performed and analysed the genetic experiments as well as the RNA binding and cleavage assays. A.W.B. set up the experiments to produce the PUCH complex from BmN4 cells, co-designed the cleavage assays and assisted in data analysis and interpretation. R.L. generated expression constructs, purified proteins for biochemical experiments and crystallization trials and performed protein interaction experiments. S.H. generated *C. elegans* strains through genome editing. E.N. and F.B. prepared samples for MS analysis, performed MS and analysed the results. T.F. purified proteins and designed and performed protein interaction experiments. E.K. performed computational analysis of the small RNA datasets. S.F. performed and interpreted AlphaFold predictions, ITC experiments and all crystallography-related work. R.F.K. assisted in data analysis and interpretation. The study was conceived and designed by S.F. and R.F.K. R.F.K., S.F., N.P. and A.W.B. contributed to writing the manuscript and making the figures, with input from all of the authors.

**Competing interests** The authors declare no competing interests.

**Additional information**
**Correspondence and requests for materials** should be addressed to Sebastian Falk or René F. Ketting.

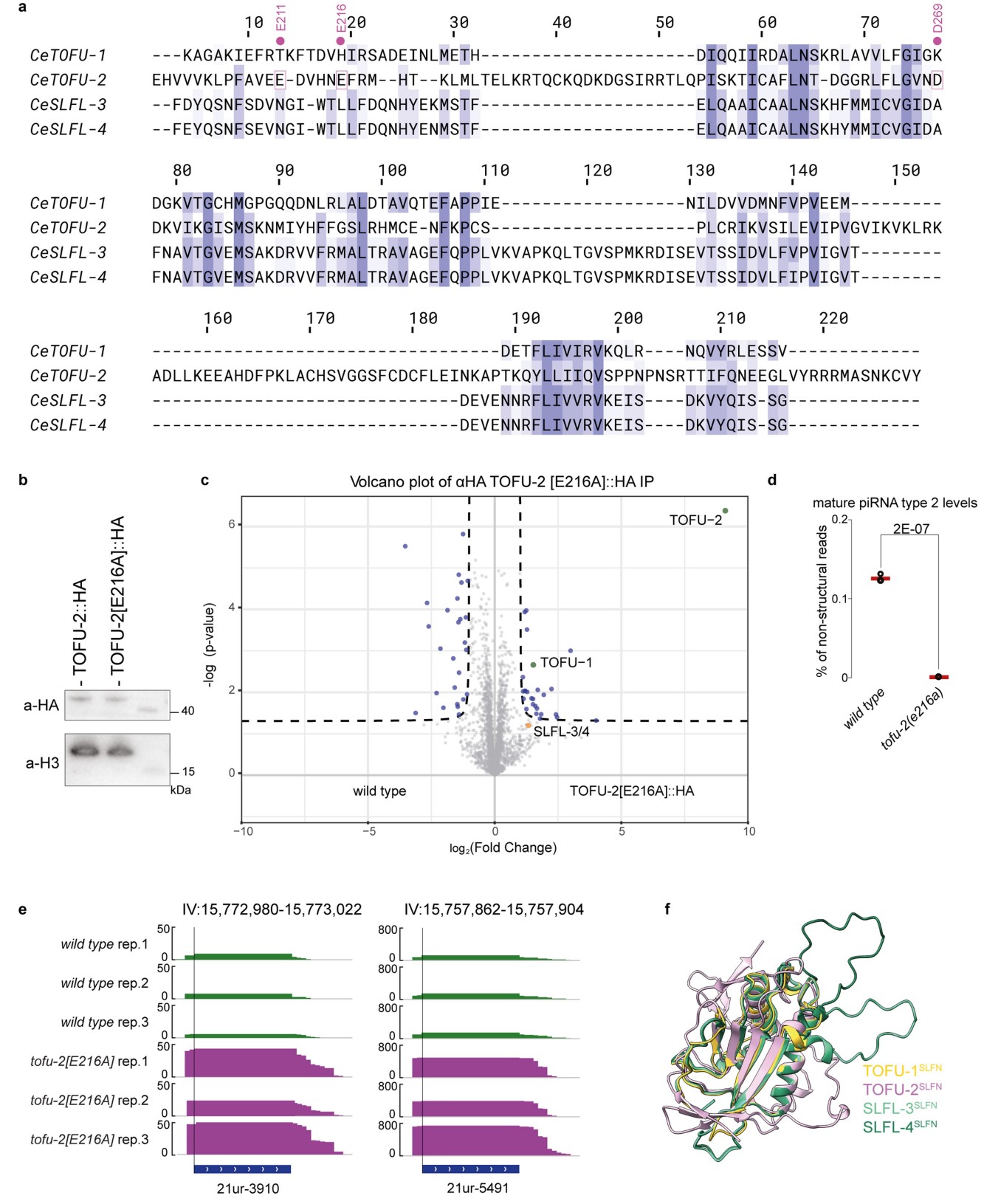

**Extended Data Fig. 1** | See next page for caption.

**Extended Data Fig. 1 | Mutation in the catalytic centre of TOFU-2 does not affect protein stability and interaction with TOFU-1. a**, Structure-based sequence alignment of the SLFN domains from TOFU-1, TOFU-2, SLFL-3 and SLFL-4. The acidic residues from the active site of TOFU-2 are highlighted with purple boxes and the residue number is indicated on the top. **b**, Extracts of young adult worms with genotype *tofu-2::HA* and *tofu-2[E216A]::HA*, were separated on SDS-PAGE. Western blot was probed using anti-HA and anti-H3 antibodies, followed by visualization with HRP. One of three experiments is shown. **c**, Volcano plot representing label-free proteomic quantification of TOFU-2[E216A]::HA and wild type immunoprecipitations from young adult extracts (n = 4 biological replicates). The X-axis represents the median fold enrichment of individual proteins in wild type (WT) versus the TOFU-2::HA mutant strain. The Y-axis indicates −log10(P-value) calculated using Welch two-sided t-test. Dashed lines represent enrichment thresholds at p-value = 0.05 and fold change > 2, c = 0.05. Each dot represents an enriched (blue/green) or quantified (grey/orange) proteins. The analysis was based on all peptides that matched to a given protein. In this experiment we could not detect unique peptides for SLFL-3 and SLFL-4. The lower enrichment of SLFL-3/4 in this experiment compared to the experiment shown in Fig. 1d most likely reflects experimental variations. PUCH stability is not affected by the TOFU-2[E216A] mutation as shown in Extended Data Fig. 5h. **d**, Total piRNA levels (type 2) in wild type and *tofu-2[E216A]*-mutant young adult hermaphrodites (n = 3 biological replicates). Red lines depict group means and P-values were calculated using two-tailed unpaired t-test. **e**, Genome browser tracks of two individual piRNA loci, displaying normalized read coverage in piRNA precursor libraries. The top three tracks (green) are from a wild type background, the bottom three tracks (purple) are from a *tofu-2[E216A]* mutant background. Note that mature piRNAs are severely depleted from precursor libraries, thus most of the depicted read coverage derive from piRNA precursors starting 2 nucleotides upstream of the 5′-ends of mature piRNAs which are indicated by a vertical line. **f**, Superposition of the AlphaFold predicted SLFN domains from TOFU-1, TOFU-2, SLFL-3 and SLFL-4. TOFU-1 is shown in yellow, TOFU-2 in purple and the paralogs SLFL-3/4 in different shades of green. Raw data are available in Supplementary Fig. 1.

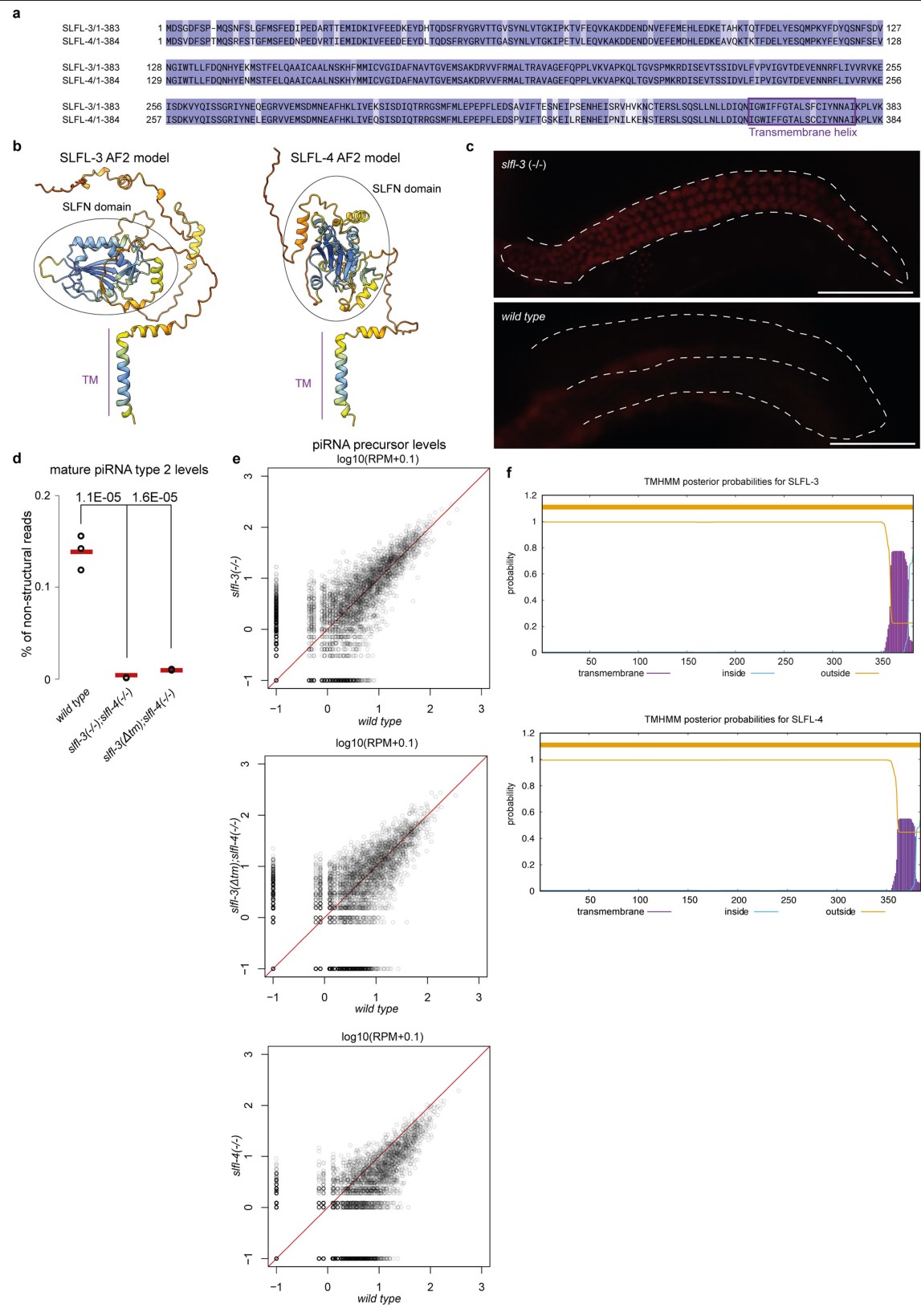

**Extended Data Fig. 2 |** See next page for caption.

**Extended Data Fig. 2 | SLFL-3 and SLFL-4 are transmembrane proteins involved in piRNA biogenesis in *C. elegans*. a**, Sequence alignment of SLFL-3 and SLFL-4. The predicted C-terminal transmembrane helix is highlighted with a box. **b**, AlphaFold2 predicted structures of SLFL-3/4 shown as cartoon and coloured by pLDDT score, which reports on the model confidence. Dark blue indicates very high, light blue confident, yellow low, and orange very low model confidence. **c**, Widefield fluorescent microscopy of adult hermaphrodites carrying the mCherry::H2B-piRNA sensor in two genetic backgrounds: *slfl-3(xf248)* on top and wild type at the bottom. The germlines are outlined by a white dashed line. Scale bar – 50 μm. Representative image from a series of 10 is shown. **d**, Total piRNA levels (type 2) in young adult hermaphrodites of the indicated genotypes (n = 3 biological replicates). Red lines depict group means and P-values were calculated using one-way ANOVA test followed by a Tukey's HSD test. **e**, Scatter plots depicting the relative abundance of type 1 piRNA precursors from individual loci in *slfl4(-/-)*, *slfl-3(ΔTM);slfl-4(-/-)* and *slfl-3(-/-)* mutants versus wild type young adult hermaphrodites (n = 3 biological replicates). RPM: Reads per million non-structural sRNA reads. **f**, Prediction of transmembrane helices in SLFL-3 and SLFL-4 using TMHMM - 2.0.

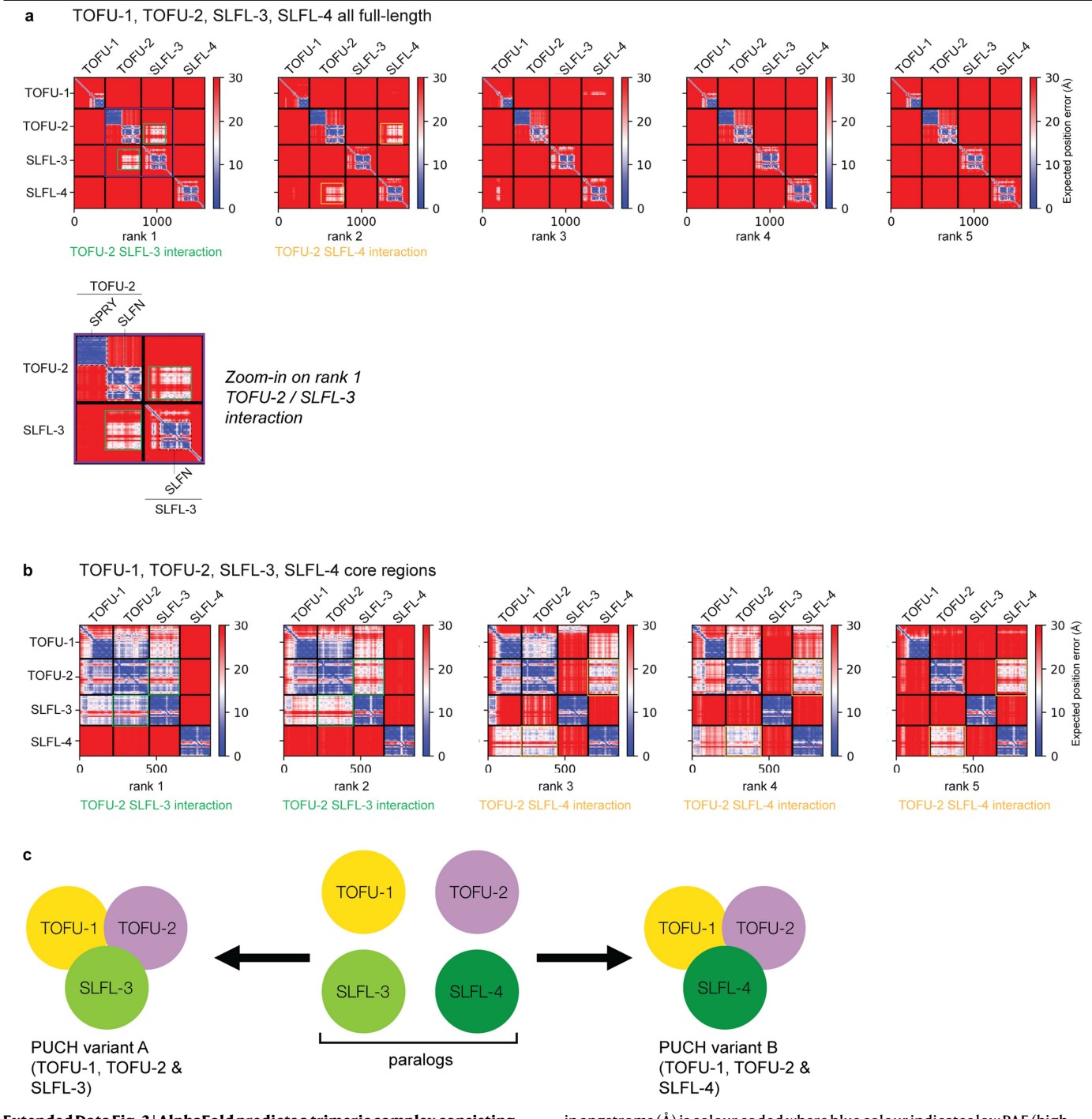

**Extended Data Fig. 3 | AlphaFold predicts a trimeric complex consisting of TOFU-1, TOFU-2 and either SLFL-3 or SLFL-4. a**, Predicted alignment error (PAE) plots for the five models predicted by Alphafold for full-length TOFU-1, TOFU-2, SLFL-3 and SLFL-4. The zoom-in highlights the predicted interaction between the SLFN domains of TOFU-2 and SLFL-3, suggesting that the TOFU-2 SPRY domain is no involved in complex formation. The expected position error in angstroms (Å) is colour coded where blue colour indicates low PAE (high confidence) and red colour indicates high PAE (low confidence). **b**, Predicted alignment error (PAE) plots for the five models predicted by Alphafold for core regions of TOFU-1, TOFU-1, SLFL-3 and SLFL-4. **c**, Schematic summary of the interaction results presented in **a** and **b**.

**a**  no template, MMSeqs2 (UniRef+Environmental), unpaired+paired, # of recycles=48

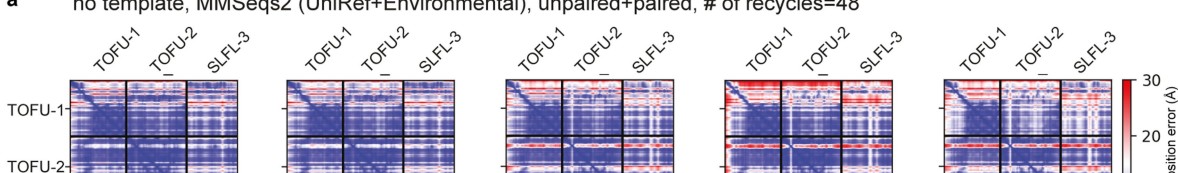

**b**  Alphafold 2 - superposition of five models

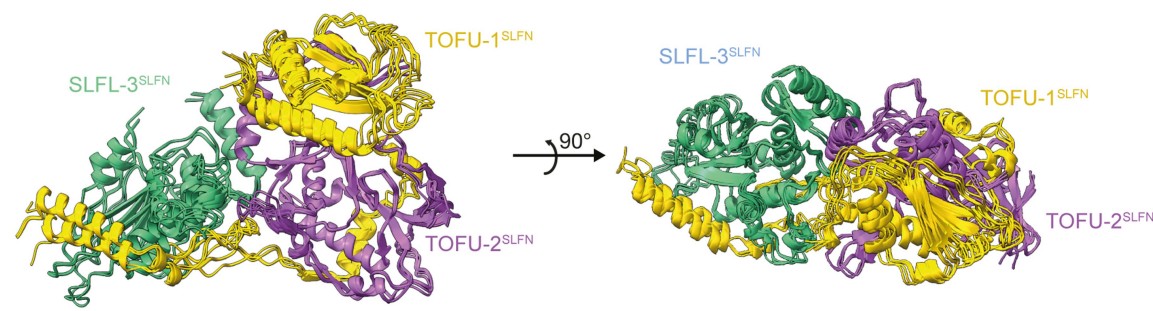

**c**  Alphafold 2 - Best model (rank 1) - colored per chain

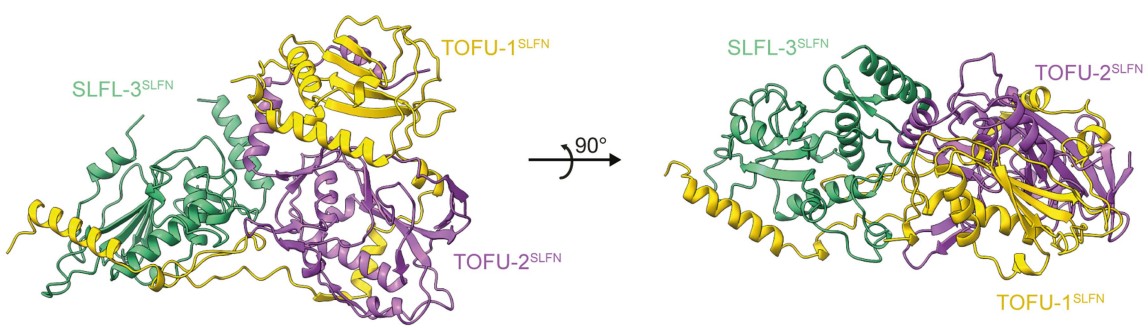

**d**  Alphafold 2 - Best model (rank 1) - colored per pLLDT

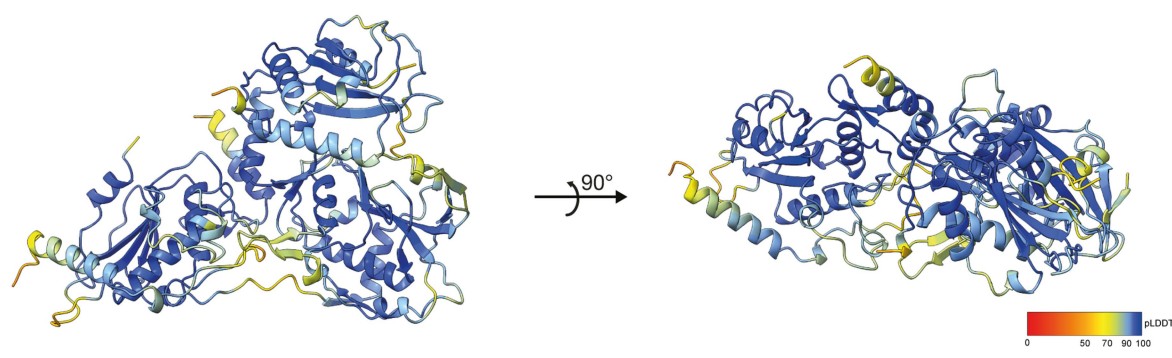

**Extended Data Fig. 4 | AlphaFold structure prediction of the trimeric complex consisting of TOFU-1, TOFU-2 and SLFL-3 shows convergence of models. a-b**, AlphaFold predicts a trimeric complex consisting of TOFU-1, TOFU-2 and SLFL-3. TOFU-1 residues 156-373, TOFU-2 residues 200-433 and SLFL-3 residues 103-300 were used for the prediction. The predicted alignment error (PAE) plots are shown in (a), the five superposed models are shown as cartoon in (b). TOFU-1 is coloured yellow, TOFU-2 purple and SLFL-3 green.

The settings used for the prediction are shown on the top. The expected position error in angstroms (Å) is colour coded where blue colour indicates low PAE (high confidence) and red colour indicates high PAE (low confidence). **c-d**, The best of the five predicted models is coloured per chain (**c**) or per pLDDT score (**d**), which reports on the model confidence. Dark blue indicates very high, light blue confident, yellow low and orange very low model confidence.

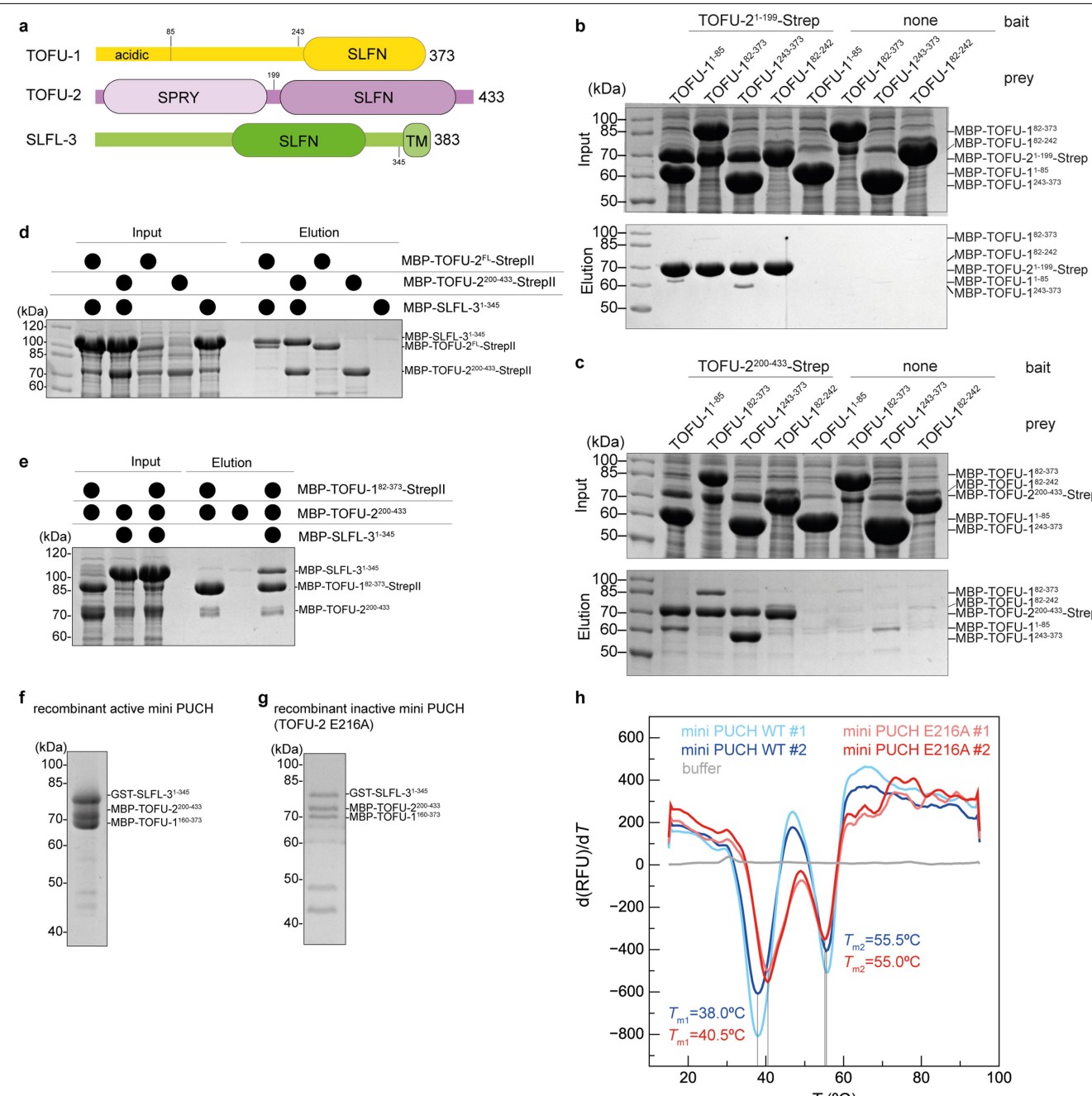

**Extended Data Fig. 5 | Verification of SLFL-domain interactions obtained by AlphaFold using recombinant proteins. a**, Schematic domain organization of TOFU-1, TOFU-2 and SLFL-3. Lines indicate low-complexity regions and rounded rectangles indicate predicted folded domains. TM: transmembrane domain. **b-c**, A construct containing the TOFU-1 SLFN domain binds to the TOFU-2 SLFN domain while the TOFU-2 SPRY domain does not bind TOFU-1. Analysis of the interaction of different TOFU-1 constructs with the StrepII-tagged TOFU-2 SPRY domain in (b) and StrepII-tagged TOFU-2 SLFN domain in (c). The indicated constructs were co-expressed in *E. coli* and the StrepII-tagged bait was precipitated by Streptactin XT beads. Input and elution fractions were analysed by SDS-PAGE followed by Coomassie staining. Experiment is done in duplicate. **d**, SLFL-3 interacts with the TOFU-2 SLFN domain. Analysis of the interaction of different StrepII-tagged TOFU-2 constructs with the SLFL-3. The indicated constructs were co-expressed in *E. coli* and the StrepII-tagged bait was precipitated by Streptactin XT beads. Input and elution fractions were

analysed by SDS-PAGE followed by Coomassie staining. Experiment is done in duplicate. **e**, TOFU-1, TOFU-2 and SLFL-3 form a trimeric complex. Different combinations of StrepII-tagged TOFU-1, TOFU-2 and SLFL-3 were co-expressed in *E. coli* and the StrepII-tagged bait was precipitated by Streptactin XT beads. Input and elution fractions were analysed by SDS-PAGE followed by Coomassie staining. Experiment is done in duplicate. **f** and **g**, Recombinant, purified mini PUCH from *E. coli* in the active form (**f**) and inactive form TOFU-2 E216A (**g**). PUCH purification was done once. **h**, The thermal stability of mini PUCH WT (shades of blue) and E216A mutant (shades of red) was assessed by differential scanning fluorimetry (DSF) for both samples in duplicates. They grey line indicates the buffer control. The first negative derivative of fluorescence intensity is plotted versus temperature. The two melting points (Tm) correspond to the two minima. RFU, relative fluorescence units. Raw data are available in Supplementary Fig. 1. Representative results are shown.

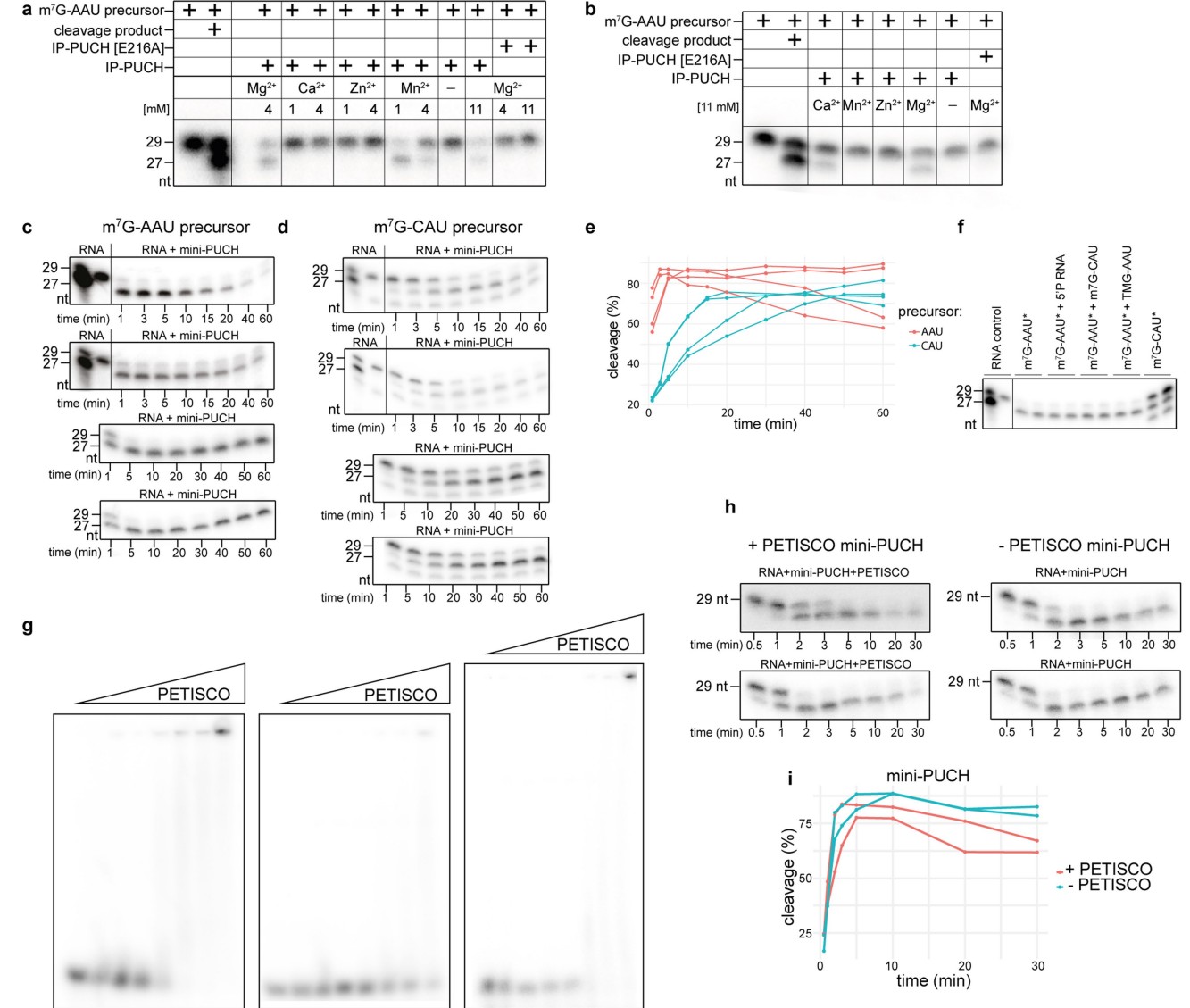

**Extended Data Fig. 6 | Characterization of PUCH activity. a-b**, Cleavage assays using GFP-IP material from BmN4 cell extracts. Beads were washed with 1 mM EDTA. Subsequently, reactions were done in buffer containing the indicated divalent cations. Concentrations were 1 or 4 mM in **a**, 11 mM in **b**. '-' indicates that no divalent cations were added during the reaction. All observations were done at least in duplicate. **c-d**, In vitro piRNA precursor cleavage assay with either m⁷G-AAU or m⁷G-CAU substrate in a time-series with recombinant mini-PUCH. Observations were made in four experiments. **e**, Quantification of the cleavage reactions, presented in **c** and **d**. **f**, Cleavage reaction with recombinant mini-PUCH of AAU substrate in the presence of different cold RNA substrates as indicated. Experiment was done in duplicate. **g**, Electrophoretic mobility shift assay between PETISCO complex and piRNA precursor with various 5'-ends. Experiment was performed twice for m⁷G-capped and 5'-OH carrying substrate and once for 5'-end substrate. **h**, In vitro piRNA precursor cleavage kinetics in presence or absence of the PETISCO complex with recombinant mini-PUCH. Experiment had been performed twice. **i**, Quantification of the cleavage reactions, presented in **h**. Raw data are available in Supplementary Fig. 1. Representative results are shown.

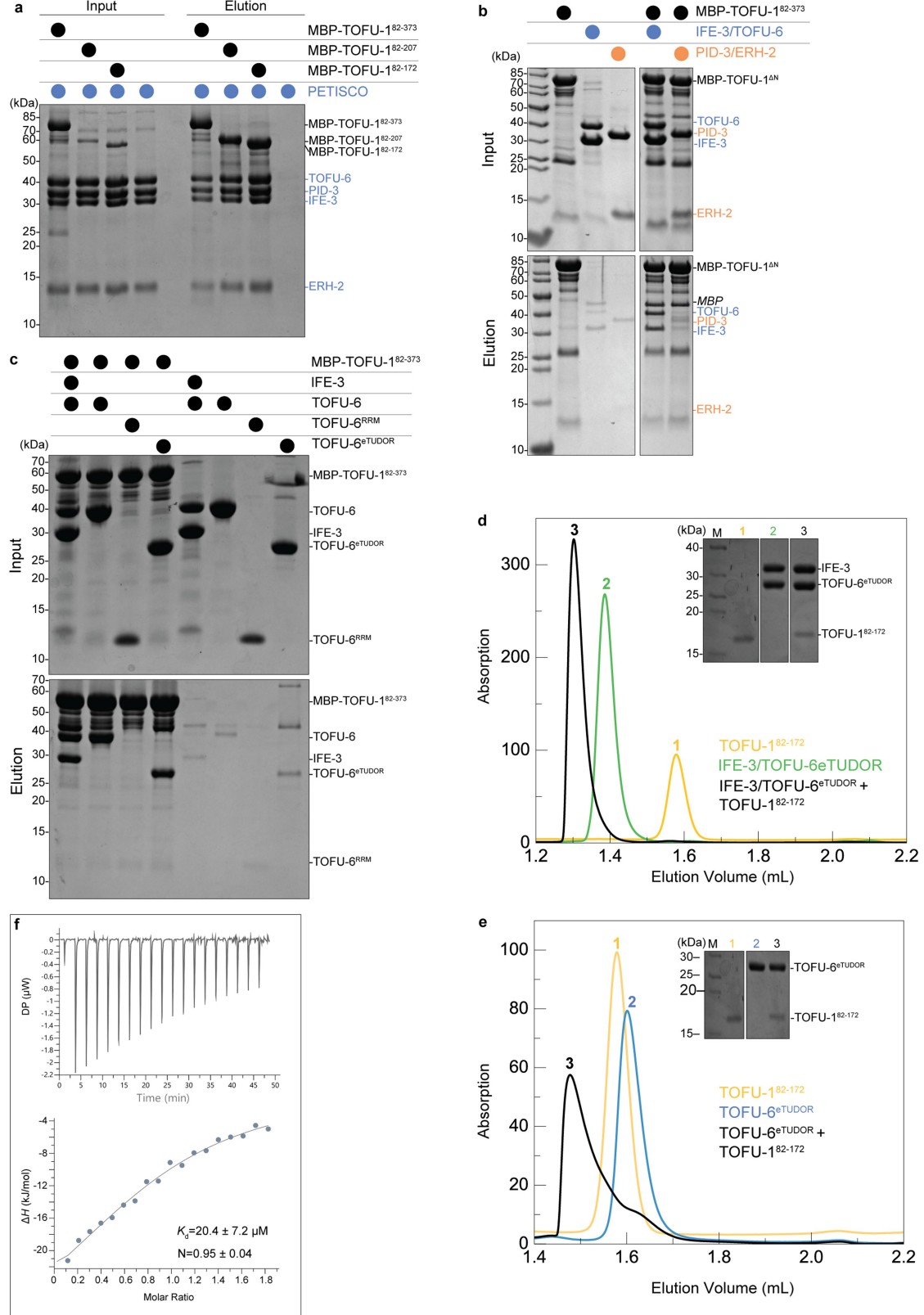

**Extended Data Fig. 7** | See next page for caption.

**Extended Data Fig. 7 | A peptide upstream of the TOFU-1 SLFN domain binds to the TOFU-6 eTUDOR domain. a-c,** Analysis of the interaction of different TOFU-1 constructs with PETISCO and its subunits by amylose pull-down assays. Input and elution fractions were analysed by SDS-PAGE followed by Coomassie staining. a, Various purified MBP-tagged TOFU-1 truncations were incubated with excess PETISCO and precipitated using amylose beads. b, Purified MBP-tagged TOFU-1$^{82-373}$ was incubated with excess of the IFE-3/TOFU-6 and PID-3/ERH-2 subcomplexes precipitated using amylose beads. c, Purified MBP-tagged TOFU-1$^{82-373}$ was incubated with excess of the IFE-3/TOFU-6 subcomplex, the TOFU-6 RRM and the TOFU-6 eTUDOR domain and precipitated using amylose beads. **d,** Purified IFE-3/TOFU-6$^{eTUDOR}$ subcomplex, TOFU-1$^{82-172}$ and a mixture thereof were subjected to size exclusion chromatography. Chromatograms: IFE-3/TOFU-6$^{eTUDOR}$ (green), TOFU-1$^{82-172}$ (yellow) and IFE-3/TOFU-6$^{eTUDOR}$ + TOFU-1$^{82-172}$ (black). The inset shows a Coomassie-stained SDS polyacrylamide gel of the peak fractions from size exclusion chromatography. **e,** Purified TOFU-6$^{eTUDOR}$, TOFU-1$^{82-172}$ and a mixture thereof were subjected to size exclusion chromatography. Chromatograms: TOFU-6$^{eTUDOR}$ (blue), TOFU-1$^{182}$ (yellow) and TOFU-6$^{eTUDOR}$ + TOFU-1$^{182}$ (black). The inset shows a Coomassie-stained SDS polyacrylamide gel of the peak fractions from size exclusion chromatography. Note: The chromatogram of TOFU-1$^{82-172}$ (yellow) is shown for comparison and is the same as shown in **d.** Also, the lanes of the polyacrylamide gel are derived from the same gel in as in **d;** thus lane 1 and the marker are identical for **d** and **e. f,** Binding of TOFU-6$^{eTUDOR}$ to TOFU-1$^{pep}$ measured by ITC. The binding affinity ($K_d$) and the stoichiometry (N) are the mean of three experiments and displayed error is the standard deviation. The experiment shows one of the three experiments as representative example. Raw data are available in Supplementary Fig. 1. Representative results are shown. All data from this figure were obtained at least in duplicate.

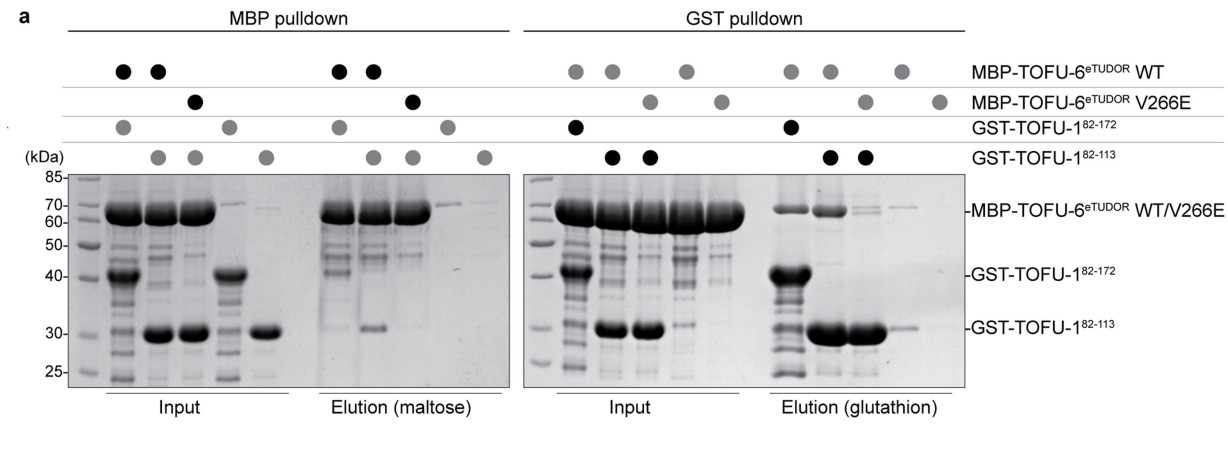

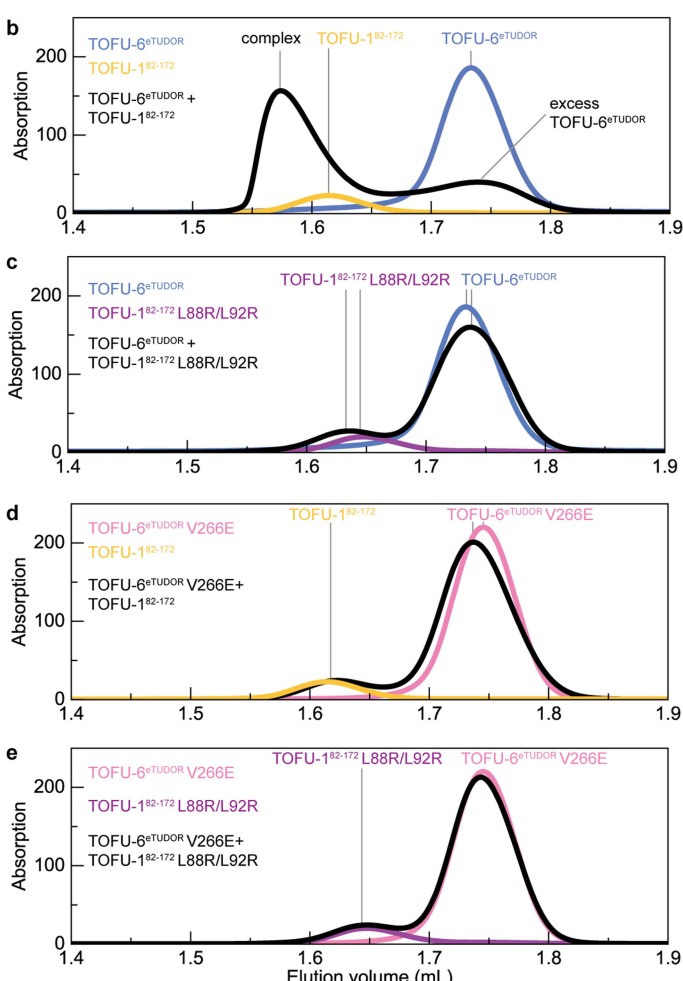

**Extended Data Fig. 8 | Structure-based analysis of the TOFU-6:TOFU-1 interaction. a**, Pull-down assays with purified recombinant wild type and mutant MBP-TOFU-6$^{eTUDOR}$ domain and GST-tagged TOFU-1$^{82-172}$ and TOFU-1$^{82-113}$ constructs. MBP pull-down assays using MBP-TOFU-6$^{eTUDOR}$ domain constructs as bait are shown on the left, GST pull-down assays using GST-tagged TOFU-1 constructs as bait are shown on right. Input and elution fractions were analysed by SDS-PAGE followed by Coomassie staining. **b-e**, Analysis of the interaction between TOFU-1 and PETISCO by size exclusion chromatography. Purified recombinant wild type and mutant versions of TOFU-6$^{eTUDOR}$ and TOFU-1$^{82-172}$ and mixtures thereof were subjected to size exclusion chromatography. Chromatograms: TOFU-6$^{eTUDOR}$ (blue), TOFU-1$^{82-172}$ (yellow), TOFU-6$^{eTUDOR}$ V266E (pink), TOFU-1$^{82-172}$ L88R/L92R (violet); the mixture of the respective proteins is always shown in black. Note: Some chromatograms are shown several times in **b-e** for direct comparison. Raw data are available in Supplementary Fig. 1. Representative results are shown. All data from this Figure was obtained at least in duplicate.

**a**

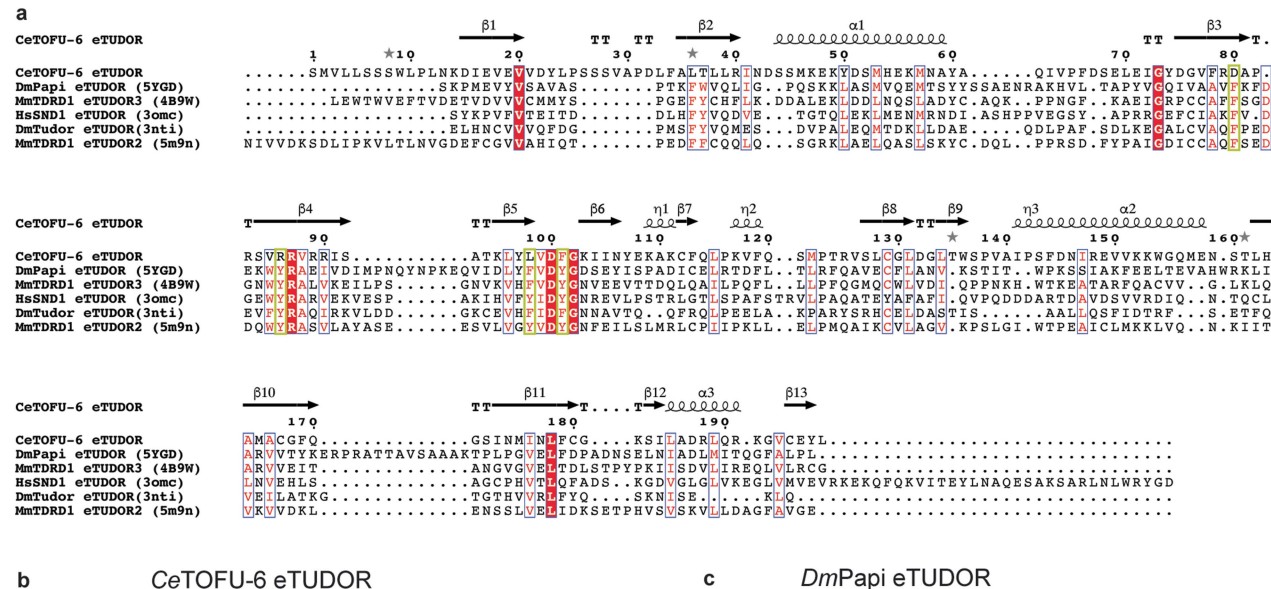

**b** *Ce*TOFU-6 eTUDOR

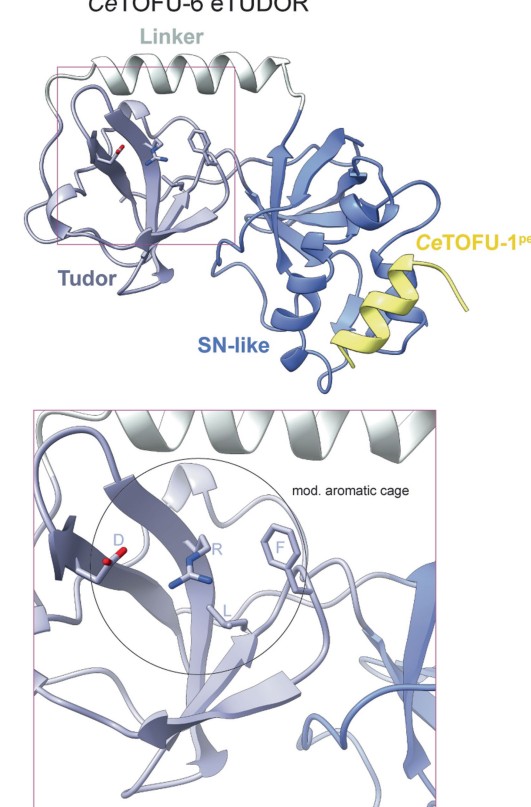

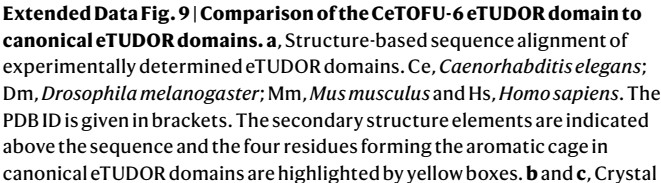

**c** *Dm*Papi eTUDOR

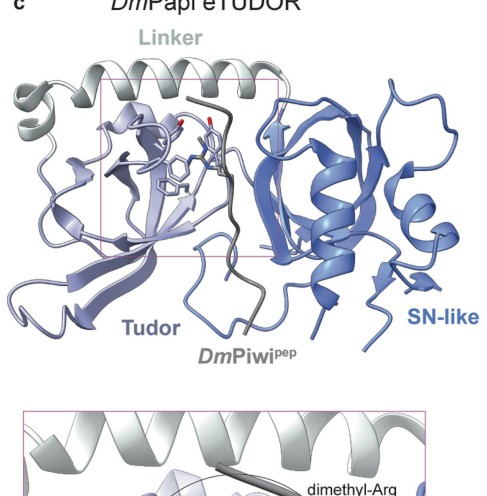

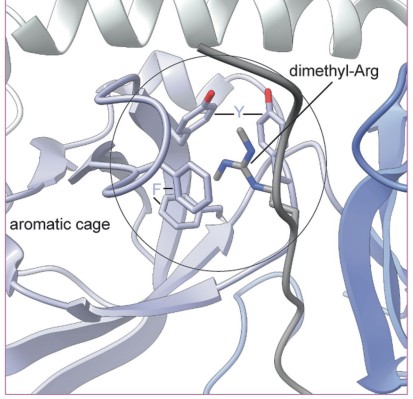

**Extended Data Fig. 9 | Comparison of the CeTOFU-6 eTUDOR domain to canonical eTUDOR domains. a**, Structure-based sequence alignment of experimentally determined eTUDOR domains. Ce, *Caenorhabditis elegans*; Dm, *Drosophila melanogaster*; Mm, *Mus musculus* and Hs, *Homo sapiens*. The PDB ID is given in brackets. The secondary structure elements are indicated above the sequence and the four residues forming the aromatic cage in canonical eTUDOR domains are highlighted by yellow boxes. **b** and **c**, Crystal structures of the *C. elegans* TOFU-6^eTUDOR–TOFU-1^pep and *D. melanogaster* PAPI^eTUDOR–PIWI^pep (PDB: 5ygd) complexes shown as cartoon. The eTUDOR domains are shown in different shades of blue, the TOFU-1^pep in yellow and the PIWI^pep containing the dimethyl-arginine residue in grey. The zoom-in view shows the region of the degenerated aromatic cage of TOFU-6^eTUDOR (**b**) and the canonical aromatic cage of PAPI^eTUDOR (**c**).

**Extended Data Table 1 | Data collection and refinement statistics**

| | TOFU-6$_{eTUDOR}$/TODU-1$^{pep}$ complex (PDB code: 9G6Z) |
|---|---|
| **Data collection** | |
| Space group | C 1 2 1 |
| Cell dimensions | |
| $a, b, c$ (Å) | 59.15, 69.20, 60.95 |
| $\alpha, \beta, \gamma$ (°) | 90.00, 91.47, 90,00 |
| Resolution (Å) | 36.52 – 2.224 (2.31 – 2.22)* |
| $R_{merge}$ | 0.11 (0.65)* |
| $I / \sigma I$ | 10.3 (2.2)* |
| Completeness (%) | 99.1 (99.2)* |
| Redundancy | 5.1 (3.6)* |
| CC (1/2) | 99.6 (77.7)* |
| | |
| **Refinement** | |
| Resolution (Å) | 2.22 |
| No. reflections | 12063 (626)* |
| $R_{work} / R_{free}$ | 0.176 / 0.230 |
| No. atoms | 1768 |
| Protein | 1679 |
| Ligand/ion | 12 |
| Water | 77 |
| $B$-factors (Å$^2$) | |
| Protein | 44.31 |
| Ligand/ion | 48.21 |
| Water | 38.55 |
| R.m.s. deviations | |
| Bond lengths (Å) | 0.004 |
| Bond angles (°) | 0.64 |

One crystal was used for structure determination. Molecular replacement was used to determine the phases. *Values in parentheses are for the highest-resolution shell.

# Reporting Summary

## Statistics

For all statistical analyses, confirm that the following items are present in the figure legend, table legend, main text, or Methods section.

| n/a | Confirmed | |
|---|---|---|
| ☐ | ☒ | The exact sample size (*n*) for each experimental group/condition, given as a discrete number and unit of measurement |
| ☐ | ☒ | A statement on whether measurements were taken from distinct samples or whether the same sample was measured repeatedly |
| ☐ | ☒ | The statistical test(s) used AND whether they are one- or two-sided<br>*Only common tests should be described solely by name; describe more complex techniques in the Methods section.* |
| ☒ | ☐ | A description of all covariates tested |
| ☒ | ☐ | A description of any assumptions or corrections, such as tests of normality and adjustment for multiple comparisons |
| ☐ | ☒ | A full description of the statistical parameters including central tendency (e.g. means) or other basic estimates (e.g. regression coefficient) AND variation (e.g. standard deviation) or associated estimates of uncertainty (e.g. confidence intervals) |
| ☐ | ☒ | For null hypothesis testing, the test statistic (e.g. *F*, *t*, *r*) with confidence intervals, effect sizes, degrees of freedom and *P* value noted<br>*Give P values as exact values whenever suitable.* |
| ☒ | ☐ | For Bayesian analysis, information on the choice of priors and Markov chain Monte Carlo settings |
| ☒ | ☐ | For hierarchical and complex designs, identification of the appropriate level for tests and full reporting of outcomes |
| ☒ | ☐ | Estimates of effect sizes (e.g. Cohen's *d*, Pearson's *r*), indicating how they were calculated |

*Our web collection on statistics for biologists contains articles on many of the points above.*

## Software and code

Policy information about availability of computer code

| Data collection | Unicorn7 software for chromatography<br>MicroCal PEAQ-ITC Control Software v1.41 to acquire ITC data<br>DM6000B microscope: LAS AF 3.1.0 8587<br>SP5 microscope: LAS AF 2.7.3.9723<br>Typhoon FLA 9500 Version V.0 Build 1.0.0.185<br>Image Lab 6.0.1 for Western blots and genotyping gels |
|---|---|
| Data analysis | The following software versions used for analysis:<br>All computational tools that were used are public and details on their use are provided in the Methods section. No custom algorithms or software were developed.<br>FastQC v.0.11.9<br>MultiQC v.1.9<br>Cutadapt v.4.0<br>Bowtie v.1.3.1<br>Samtools v.1.10<br>GNU Awk v.5.1.0<br>Subread v.1.6.2<br>Bedtools v.2.27.1<br>kentUtils v.385<br>IGV v.2.15.4<br>R v.4.1.0  and v5 |

```
Phenix v.1.20.1-4487
ChimeraX v.1.5
autoPROC v.1.1.7
XDS VERSION Feb 5, 2021
AIMLESS v. 0.7.7
COOT v0.9.8.6
MaxQuant suite 1.6.5.0
Adobe Illustrator 2023
ImageJ 64 V5
Datagraph5
```

For manuscripts utilizing custom algorithms or software that are central to the research but not yet described in published literature, software must be made available to editors and reviewers. We strongly encourage code deposition in a community repository (e.g. GitHub). See the Nature Portfolio guidelines for submitting code & software for further information.

## Data

Policy information about availability of data

All manuscripts must include a data availability statement. This statement should provide the following information, where applicable:
- Accession codes, unique identifiers, or web links for publicly available datasets
- A description of any restrictions on data availability
- For clinical datasets or third party data, please ensure that the statement adheres to our policy

Sequencing data is available at NCBI's Sequence Read Archive under accession number PRJNA925182 (https://dataview.ncbi.nlm.nih.gov/object/PRJNA925182?reviewer=951lavsn8a0umpj5j17m8bk738).
The mass spectrometry proteomics data have been deposited to the ProteomeXchange Consortium via the PRIDE72 partner repository with the dataset identifier PXD039502. Username: reviewer_pxd039502@ebi.ac.uk
Password: T2C6anBX
Coordinates and structure factors of the TOFU-6eTUDOR TOFU-1pep complex structure have been deposited in the Protein Data Bank with accession codes PDB ID 8BY5.
Wormbase WS289 was used in this study.
Uniprot was regularly used. Because of constant updates specific versions cannot be given. Always the most recent version was used.
Alphafold database (https://alphafold.ebi.ac.uk/) was used in this work.

## Human research participants

Policy information about studies involving human research participants and Sex and Gender in Research.

| | |
|---|---|
| Reporting on sex and gender | NA |
| Population characteristics | NA |
| Recruitment | NA |
| Ethics oversight | NA |

Note that full information on the approval of the study protocol must also be provided in the manuscript.

# Field-specific reporting

Please select the one below that is the best fit for your research. If you are not sure, read the appropriate sections before making your selection.

☒ Life sciences　　☐ Behavioural & social sciences　　☐ Ecological, evolutionary & environmental sciences

For a reference copy of the document with all sections, see nature.com/documents/nr-reporting-summary-flat.pdf

# Life sciences study design

All studies must disclose on these points even when the disclosure is negative.

| | |
|---|---|
| Sample size | Sample sizes for the worm experiments were not based on statistical methods but on previously published similar experiments yielding consistent and reproducible results. For small RNA sequencing 3 biological replicates yield very robust results.<br>Sample sizes for mass spectrometry experiments were established as quadruplicates (LFQ), as is well-accepted in the proteomics field. The four samples were biological replicates. For immunoprecipitation and Western Blot experiments, animals were prepared from bleaching 1-2 high density plates of gravid adults per sample, L4 samples, young adult samples, and gravid adult samples were prepared from a pool of synchronized animals (200 µl per sample). All blots and IPs were done in two independent biological experiments. |
| Data exclusions | No data was excluded |

| Replication | All gels and blots were done in duplicate.<br>All protein analyses were done in duplicate.<br>Sequencing data were obtained from one experiment based on biological triplicates. In our experience this yields very robust results. Many papers build on duplicates only. The tofu-2(e216a) mutant was analyzed twice in this manner.<br>Mass spectrometry is based on one experiment that uses biological quadruplicates. This is widely accepted practice in the field of quantitative proteomics.<br>Imaging results were based on observations on different individuals animals. At least ten animals were imaged and a representative image was used in the manuscript. Images of cells were obtained with two different transfection experiments. Representative images of cells are shown in the manuscript. Seven cells were imaged in both experiments.<br>All PUCH cleavage reactions were done in duplicate. The ligation experiment was done in triplicate. Cleavage of the 10 nucleotide substrate and the gel-shift with the 5'P end were the only experiments performed once.<br>In all cases replication yielded equivalent results. |
|---|---|
| Randomization | Randomization is not relevant to this study, as distinct genotypes and protein preparations had to be generated before each experiment. |
| Blinding | Blinding was used in the initial analysis of the sRNA data. The bio-informatician received only strain names without genotype information. After initial analysis, results were interpreted in light of the genotypes, so at this stage no blinding was applied. This is also not possible if questions are derived from the initial results of analysis.<br>Blinding was applied in the in vitro PUCH activity assays from lysates of transfected BmN4 cells. Lysate sources were unknown to the N. Podvalnaya when running the cleavage assays.<br>In all other experiments blinding was not used in experiments for the following reasons:<br>The primary results of our work were rather objective, making blinding not needed.<br>The explorative nature of the study makes blinding impossible and rather unlikely to affect the results.<br>Genotypes of animals needed to be established before analysis. |

# Reporting for specific materials, systems and methods

We require information from authors about some types of materials, experimental systems and methods used in many studies. Here, indicate whether each material, system or method listed is relevant to your study. If you are not sure if a list item applies to your research, read the appropriate section before selecting a response.

## Materials & experimental systems

| n/a | Involved in the study |
|---|---|
| ☐ | ☒ Antibodies |
| ☐ | ☒ Eukaryotic cell lines |
| ☒ | ☐ Palaeontology and archaeology |
| ☐ | ☒ Animals and other organisms |
| ☒ | ☐ Clinical data |
| ☒ | ☐ Dual use research of concern |

## Methods

| n/a | Involved in the study |
|---|---|
| ☒ | ☐ ChIP-seq |
| ☒ | ☐ Flow cytometry |
| ☒ | ☐ MRI-based neuroimaging |

## Antibodies

| Antibodies used | Monoclonal anti-HA (clone 12CA5) mouse antibody, in-house production; 1:1,000<br>anti-Histone H3, Art. No. H0164, Sigma-Aldrich; 1:1,000<br>anti-mouse IgG, HRP-linked antibody, Art. No. 7076, Cell Signaling Technology; 1:10,000<br>anti-rabbit IgG, HRP-linked antibody, Art. No. 7074, Cell Signaling Technology; 1:10,000<br>Monoclonal anti-MYC (clone 9B11) mouse antibody, #2276S, Cell Signaling; 1:1,000<br>Anti-Actin polyclonal antibody raised in Rabbit (Sigma, #A5060); 1:1,000<br>IRDye® 800CW Goat anti-Mouse IgG Secondary Antibody (LI-COR, #926-32210); 1:10,000<br>IRDye® 680LT Donkey anti-Rabbit IgG (LI-COR, #926-68023); 1:10,000<br>Monoclonal ANTI-FLAG® (clone M2), Art. No. F3165, Sigma-Aldrich; 1:1,000<br>Monoclonal anti-HA (clone 12CA5) mouse antibody, in-house production; 1:1,000<br>Monoclonal Anti-GFP Antibody (clone B-2), Santa Cruz, Cat. #sc-9996, Lot.#K1115; 1:1,000 |
|---|---|
| Validation | BmN4 cell work:<br>Absence of signal in absence of expression of tagged protein was taken as verification of specificity in all cases.<br>Anti-Actin (Sigma, #A5060). Quality control by Sigma-Aldrich: working dilutions for western blot of at least 1:250 were determined using rat brain or chicken muscle extracts<br><br>IRDye® 800CW Goat anti-Mouse IgG Secondary Antibody (LI-COR, #926-32210): Isolation of specific antibodies was accomplished by affinity chromatography using pooled mouse IgG covalently linked to agarose. Based on ELISA and flow cytometry, this antibody reacts with the heavy and light chains of mouse IgG1, IgG2a, IgG2b, and IgG3, and with the light chains of mouse IgM and IgA. This antibody was tested by dot blot and and/or solid-phase adsorbed for minimal cross-reactivity with human, rabbit, goat, rat, and horse serum proteins, but may cross-react with immunoglobulins from other species. The conjugate has been specifically tested and qualified for Western blot applications. |

IRDye® 680LT Donkey anti-Rabbit IgG (LI-COR, #926-68023). The antibody was isolated from antisera by immunoaffinity chromatography using antigens coupled to agarose beads. Based on immune electrophoresis, this antibody reacts with the heavy chains of rabbit IgG, and with the light chains common to most rabbit immunoglobulins. No reactivity was detected against non-immunoglobulin serum proteins. This antibody was tested by ELISA and/or solid-phase adsorbed to ensure minimal cross-reactivity with bovine, chicken, goat, guinea pig, Syrian hamster, horse, human, mouse, rat, and sheep serum proteins, but may cross-react with immunoglobulins from other species. The conjugate has been specifically tested and qualified for Western blot applications.

Worm work:
Specificity tested by absence of signal in lysates from untagged strains.
Monoclonal anti-HA (clone 12CA5) antibody: Soluble lysate from exponentially-growing yeast cultures expressing HA-tagged proteins were separated on 4-15% gradient gel (BioRad). Proteins were blotted an a nitrocellulose membrane using semi-dry transfer (10 min High MW program on TransBlot Turbo blotter) and the membrane was blocked 1h in 5% skim milk/PBS/0.1% Tween-20. Primary antibodies: mouse anti-HA (in-house or Covance, 1:1000 o/N in blocking solution) at 4°C Secondary antibody: goat anti-mouse HRP-coupled (BIORAD 170-5047) 1:3000 1h at RT in blocking solution. The western blot was developed using ECL substrate Dura (Thermo, #34076). Exposure times as indicated in the figure. The experiment was performed by Katharina Bender from Brian Luke's group, IMB, Mainz, Germany.
 anti-Histone H3 antibody: By immunoblotting, a working antibody dilution of 1:5,000-1:10,000 is recommended using a whole cell extract of the A431 human epidermoid carcinoma cell line, and a whole cell extract of the mouse fibroblast NIH3T3 cell line. By immunoblotting, a working antibody dilution of 1:2,500-1:5,000 is recommended using a whole cell extract of the rat pheochromocytoma PC12 cell line.
 anti-mouse IgG, HRP-linked antibody: Application Key: WB-Western Blot IP-Immunoprecipitation IHC-Immunohistochemistry ChIP-Chromatin Immunoprecipitation IF-Immunofluorescence F-Flow Cytometry E-P-ELISA-Peptide. Species Cross-Reactivity Key: H-Human M-Mouse R-Rat Hm-Hamster Mk-Monkey Vir-Virus Mi-Mink C-Chicken Dm-D. melanogaster X-Xenopus Z-Zebrafish B-Bovine Dg-Dog Pg-Pig Sc-S. cerevisiae Ce-C. elegans Hr-Horse All-All Species Expected  anti-rabbit IgG, HRP-linked antibody: Application Key: WB-Western Blot IP-Immunoprecipitation IHC-Immunohistochemistry ChIP-Chromatin Immunoprecipitation IF-Immunofluorescence F-Flow Cytometry E-P-ELISA-Peptide. Species Cross-Reactivity Key: H-Human M-Mouse R-Rat Hm-Hamster Mk-Monkey Vir-Virus Mi-Mink C-Chicken Dm-D. melanogaster X-Xenopus Z-Zebrafish B-Bovine Dg-Dog Pg-Pig Sc-S. cerevisiae Ce-C. elegans Hr-Horse All-All Species Expected
Monoclonal anti-MYC: no signal in non-tagged C. elegans strains; signal at expected MW in tagged strains only.

# Eukaryotic cell lines

Policy information about cell lines and Sex and Gender in Research

| | |
|---|---|
| Cell line source(s) | BmN4 cells used in this study were provided to us by Dr. Ramesh Pillai, University of Geneva, in 2015. BmN4 cells originate from Kyushu University, Dr. Kusakabe. Also see: https://www.cellosaurus.org/CVCL_Z634 |
| Authentication | BmN4 cells have not been authenticated. |
| Mycoplasma contamination | Not tested. |
| Commonly misidentified lines (See ICLAC register) | No commonly misidentified cell lines were used. |

# Animals and other research organisms

Policy information about studies involving animals; ARRIVE guidelines recommended for reporting animal research, and Sex and Gender in Research

| | |
|---|---|
| Laboratory animals | In all experiments young adult animals were used.<br>Caenorhabditis elegans strains were all based on the wild-type isolated Bristol N2, which was used as control in many experiments.<br>Strains developed and/or used in this study:<br>RFK1269 tofu-2(xf245[tofu-2::HA]), V.<br>RFK1273 tofu-2(xf246[E216A]::HA]), V.<br>RFK1059 tofu-2(xf231[E216A]]) V.<br>RFK1242 pid-1(xf35) II; tofu-2(xf231) V.<br>RFK1095 mjSi22 [Pmex-5::mCherry::his-58::21UR-1_as::tbb-2(3'UTR)] I, tofu-2(xf231) V.<br>RFK1246 mjSi22 [mex-5p::mCherry::his-58 + 21UR-1_as + tbb-2(3'UTR)] I; mut-7(xf125) III.<br>RFK851 mjSi22 [Pmex-5::mCherry::his-58::21UR-1_as::tbb-2(3'UTR)] I prg-1(n4357) I.<br>RFK204 mjSi22 [Pmex-5::mCherry::his-58::21UR-1_as::tbb-2(3'UTR)] I.<br>RFK1481 slfl-3(xf248) I.<br>RFK1580 tofu-1(xf337[L88R;L92R]) V.<br>RFK1506 tofu-6(xf312[V266E]::3xMYC) I.<br>RFK1605 tofu-1(xf337) V; tofu-6(xf312) I.<br>RFK1057 tofu-6(xf229[tofu-6::3xMYC]) II.<br>RFK1639 slfl-4(xf351) IV.<br>RFK1640 slfl-3(xf248) I, slfl-4(xf351) IV.<br>RFK1689 slfl-3(xf356) I; slfl-4(xf351) IV.<br>RFK1692 tofu-1(xf358[tofu-1::3MYC]), V.<br>RFK1693 tofu-1(xf363[3MYC::tofu-1[L88R&L92R]]) V. |
| Wild animals | No wild animals were used in this study |

| | |
|---|---|
| Reporting on sex | All studies used hermaphrodites. Males occur spontaneously in cultures, but only at a low frequency, and for these studies can be neglected. |
| Field-collected samples | No filed-collected samples were used in this study |
| Ethics oversight | This study did nor require ethical approval |

Note that full information on the approval of the study protocol must also be provided in the manuscript.

