## [Peer Review File · Nature]

Manuscript Title: piRNA processing by a trimeric Schlafen-domain nuclease

Reviewer Comments & Author Rebuttals

Reviewer Reports on the Initial Version:

Referees' comments:

Referee #1 (Remarks to the Author):

In this manuscript, Ketting and colleagues identified an endonuclease called "PUCH" that produces the 5' end of mature piRNAs from nascent piRNA precursors in *C. elegans*. PUCH consists of three Schlafen-like-domain proteins (SLFL proteins), TOFU1 (SLFL-1), TOFU2 (SLFL-2) and SLFL-3 or 4. By wisely combining structure prediction by AlphaFold2 and functional validation by biochemical and genetic experiments, the authors demonstrate the catalytic mechanism and substrate specificity of PUCH. Moreover, they showed how PUCH interacts with PETISCO for efficient processing of piRNA precursors. Although this PUCH-mediated piRNA-processing mechanism per se is likely conserved only within worms, the current findings make important contributions to the piRNA field and also open the door to the new concept of multimeric Schlafen-domain endonucleases even in other species.

Major Comments:

1. The authors show that PUCH strictly recognizes the m7G cap and make an intriguing proposal that this substrate specificity distinguishes piRNA precursors from mRNAs, which have trimethyl G (TMG) cap in *C. elegans*. Directly testing this model by using a TMG-capped AAU precursor in their cell free systems will significantly strengthen this study. Also, the authors should discuss if there can be any other m7G-capped transcripts in worms, and if so how PUCH (and PETISCO) acts on them.
2. The conserved "YR" motif is known to be important for transcription of piRNA precursors in worms. Do these two nucleotides (right upstream of the third "U") affect the processing by PUCH? Also, does PUCH have any preference for the length of substrate RNAs?
3. As currently discussed in the manuscript, it will be important to experimentally examine the functional relevance of the transmembrane helix of SLFL-3 (and/or 4) and thereby the mitochondrial localization of PUCH in vivo.
4. In Fig. S1e, mature piRNAs levels appear to be comparable with their precursors in wildtype. In contrast, Fig. 5d and e show that overall mature piRNAs levels are >10 fold more abundant than

their precursors in wildtype. Does this mean that the two piRNAs shown in Fig. S1e are peculiar examples?

5. Related to the above point, the current sequencing analysis relies on a very stringent definition of mature piRNAs and their precursors (pages 14–15), which may explain the apparent discrepancy above. The authors should present evidence to support the appropriateness of the definitions. Also, the authors should show how much fraction of non-structural reads are mapped to piRNA loci outside of mature piRNAs or piRNA precursors.

Minor Comments:

1. The statement “The identification of PUCH completes the repertoire of *C. elegans* piRNA biogenesis factors.” in Abstract may be too strong, as there may be other unknown piRNA biogenesis factors.
2. Fig. 1d shows much more SLFL-3 than SLFL-4 in TOFU-2 IP, whereas Fig. S1d shows apparently no SLFL-3 in TOFU-2[E216A] IP. The authors should give a reasonable explanation for this.
3. In Fig. S1e, addition of vertical lines to indicate the 5' end of mature piRNAs will be helpful.
4. The overall upper shift in Fig. 1h etc. can be explained by the accumulation of piRNA precursors, the reduction of mature piRNAs, or both. An internal control will be helpful to distinguish these possibilities.
5. In Fig. 4b, does the shown p-value refer to *tofu-2*[E216A]-mutant or *tofu-2*[E216A];*pid-1*(xf35)?
6. Quantification and reproducibility should be shown in Fig. 4c.
7. The right panel in Fig. 5f lacks “1” of “*tofu-1*.” Also, *tofu-1*[L88R;L92R];*tofu-6*[V266E] may be better compared with *tofu-6*[V266E] rather than the wildtype.
8. “SLFL3/4” in Fig. S5a should read just “SLFL3.”
9. Page 41, legend of Fig. S7d and e: “TOFU-1_72-182” and “TOFU-1_182” should probably read “TOFU-1_82-172” (as in the Figures).
10. After processing by PUCH, the resultant 5'-U piRNA precursors are expected to be loaded into PRG-1. Do the authors know if any components of PUCH interact with PRG-1?
11. “Piwi” can mean a specific PIWI protein (among Aub, Ago3, and Piwi) in *Drosophila*. “PIWI” should instead be used when referring to the protein family to avoid any confusion.

Referee #2 (Remarks to the Author):

A. Summary of the key results

This manuscript by Podvalnaya and coworkers focuses on the mechanisms involved in the maturation of piRNAs in *C. elegans* worm as model system. The piRNAs are small non-coding RNAs involved in the formation of the RNA-induced silencing complex to post-transcriptionally silence transposable elements and contribute to the regulation of gene expression. In *C. elegans*, these piRNAs are transcribed as precursors harboring a 5' cap and upon the action of several enzymes or multi-protein complexes, these precursors are matured into 21 nucleotides piRNAs (21U piRNAs), all of which have a U at their 5' end.

The piRNA biogenesis process is highly complex in eukaryotes and in *C. elegans*, it relies on the action of the PETISCO complex, on PARN-1 and HENN-1 enzymes for 3' end trimming and methylation, respectively. As the machinery responsible for the cleavage of the 5' cap to generate 21U piRNAs is still unknown, the authors have used a multidisciplinary approach to identify a protein

complex as essential for this process. Building up on the observation that TOFU-1 and TOFU-2 proteins are necessary for piRNA accumulation, the authors have used mass spectrometry as well as recombinant proteins (expressed either in insect cells or in *E. coli*) to identify a trimeric complex (dubbed PUCH) formed by TOFU-1, TOFU-2 and SLFL-3/4 proteins as the enzyme involved in the maturation of precursors piRNAs into 21U piRNAs. They also perform *in vivo* experiments to evaluate the effect of point mutants on the generation of 21U piRNAs. Finally, they demonstrate an interaction between PUCH and PETISCO and have determined the structure of the complex between the TOFU-1 subunit of the PECH complex and the TOFU-6 subunit from PETISCO. The authors also elegantly take advantage of the recently generated structure prediction of the isolated proteins thanks to the AlphaFold2 program and use the AlphaFold-multimer version to generate models of the trimeric PECH complex.

B. Originality and significance:

In eukaryotes, the piRNAs play crucial roles in targeting the Argonaute proteins to other RNAs to induce their silencing. Their biogenesis process and roles have been scrutinized since their discovery using mouse and fruit-fly as model systems. This field has been the subject of publications in very high impact journals. This is particularly true for studies focusing on mouse and *D. melanogaster* piRNA maturation and function. In *C. elegans*, the biogenesis pathway of 21U piRNAs differs compared to the two model organisms mentioned above. The originality of this study is that it contributes to filling the last gap in the complex mechanism of piRNA maturation in *C. elegans* by identifying the PUCH complex as responsible for the endonucleolytic processing of precursor piRNAs to generate 21U piRNAs. This is then of great interest but as this specific mechanism does not exist in mammals, this is reducing the broad interest of this study from my point of view. Based on these results, it could have been interesting to discuss more deeply the potential roles of SLFL proteins present in mammals.

C. Data & methodology: validity of approach, quality of data, quality of presentation

Overall, this manuscript is of high quality and the methods implemented are well adapted to address the problem. Yet, additional controls are needed for some experiments to fully support the conclusions.

Major points to be addressed :

- Figure 2, panel a : Lanes 5 to 7 and to a lesser extent lane 4. The amount of eGFP-TOFU-2-WT used in these pull-down experiments differs compared to the other lanes. This is a problem as it is not possible to compare these lanes (which consist mostly in controls) with the other ones.
- Figure 3, panel b. In this experiment, endonuclease activity is only detected when the three wild-type proteins (TOFU-1, TOFU-2 and SLS-3) are present. The authors show a control with the TOFU-2 E216A mutant in the presence of TOFU-1. In this context, they do not detect enzymatic activity. However, as the complex between TOFU-1 and TOFU-2 WT does not exhibit enzymatic activity, this control cannot allow concluding that the TOFU-2 is the endonuclease and that E216 is a mandatory residue. A control of the TOFU-2 E216A mutant in the presence of both TOFU-1 and SLSL-3 proteins is needed. Consequently, the conclusion « Introduction of an E216A mutation in TOFU-2 completely blocked this reaction (Fig. 3b). » (page 6, lanes 26-27) is not supported by the presented experiment.
- For several figures, the indication of the components used in the corresponding experiments is confusing. For instance, Fig. 3c, lanes 4 and 5 both contain cap-AAC precursor and recombinant WT PUCH. Is this correct? The same is true for lanes 6 and 7 of the same panel. Fig S7b, lanes 3 and 4 :

MBP-TOFU-1-(82-373) is wrongly indicated to be present in these conditions.

- Figure 3d, lane 3. The authors seem to have omitted to mention that the cap-AAU precursor is present in lanes 3 and 4. In lane 3, the intensity of the band corresponding to the expected cleaved RNA ligated to the 10 nucleotide adapter is weak and the bands corresponding to the cap-AAU precursor and the cleavage product have higher intensity. This should be improved for better readability.
- Figure 5e. It is crucial to validate that the different TOFU-1 and TOFU-6 mutants are well expressed in the cells used for these experiments. Without this control, it is not possible to conclude that the observed effect on levels of mature and precursor piRNAs are indeed due to the disruption of the interaction between TOFU-1 and TOFU-6.

Minor points to be addressed :

- The TOFU-2 E216A mutant seems to be less soluble than wild-type (Fig. S5g) and to interact more weakly with TOFU-1 and SLFL-3 (Fig. S1d). This should be discussed and in particular in light of the location of this residue in the model of the PUCH complex.
- Page 6, lane 29. It is stated that the precursor processing activity is inhibited by EDTA. I have scrutinized the material and methods and all the text but I did not find any detail about EDTA concentration. It should be clearly indicated in Fig S6a-b, which lanes correspond to the experimental conditions with EDTA. I guess those labeled with a « - » sign but it should be clarified.
- Page 8, lanes 30-31 : It is not clear how the authors narrowed down the TOFU-1 region interacting with PETISCO from residues 82-172 to 82-113. This should be explained.
- Fig.1b. Please specify from which organism SLFN13 originates.
- Fig. S6b. It is stated in the materials and methods that the metal concentrations used in this experiment are 100 mM, which is radically different from the concentrations indicated on this panel. Please clarify.
- Figure S9a : Due to the darkness, the details of this gel are difficult to see.
- Figure S9c-e : The OD values for the Y-axes of these curves seem to have been normalized. From my point of view, this is quite confusing as one protein of the complex seems to be present in lower amount (differences in the height of the peak for TOFU-1 proteins alone or incubated with TOFU-6). I would suggest to use the measured OD values. This should facilitate the comparison between the different curves.

Typos to be corrected :

- Page 14 lane 15 and page 17 lane 8 : TRISOL should be TRIzol
- Page 14, lane 17 : 21xg seems to be too low. Please check.
- Page 19, lane 17 : Supdersex should be Superdex
- Page 19, lane 19 : Crystallization should be crystallization.

D. Appropriate use of statistics and treatment of uncertainties

The statistical tests are appropriate.

E. Conclusions: robustness, validity, reliability

Some conclusions are not fully supported by the presented experiments.

F. Suggested improvements: experiments, data for possible revision

In addition to the above mentioned major points, it would be interesting to indicate the location of E216 residues from TOFU-2 in the model of the TOFU-1/TOFU-2/SLFL-3 structure presented in Figure 2c.

G. References: appropriate credit to previous work?

The references in the field are adequately cited.

H. Clarity and context: lucidity of abstract/summary, appropriateness of abstract, introduction and conclusions

The manuscript is clearly written although as mentioned above some conclusions are not fully supported by the presented experiments.

Referee #3 (Remarks to the Author):

The complexity of small RNA pathways in *C. elegans* is an endless source of new biological discoveries, and a fascinating new one is described by the authors of this manuscript, concerning the elusive nuclease complex that produces the 5' ends of *C. elegans* piRNAs (21U RNAs).

In *C. elegans*, individual piRNA precursors are transcribed by Pol II and contain a 5'-cap; they are further cleaved by an unknown endonuclease, which generates the mature 5' end of the piRNA that will be loaded to PRG1 and further trimmed and 2-O' methylated to generate functional piRNAs. The authors have previously discovered the PETISCO complex (composed of PID-3, ERH-2, TOFU-6, IFE-3) as essential for piRNA biogenesis. However, PETISCO did not exhibit any nucleolytic activity and its role is most likely to recognize and stabilize piRNA precursors.

In their latest, tour de force study, the authors identify the elusive worm piRNA endonuclease and make the surprising discovery that it resides in a complex of four novel, Schlafen-like (SLFN) proteins that interact as a heterotrimer, which they call PUCH, which catalyzes piRNA cleavage in a cap a sequence depended manner, ensuring accurate piRNA processing. They show that PUCH interacts with PETISCO, via a short segment of TOFU-1, one of the SLFN-like proteins, with the extended Tudor (eTUD) domain of TOFU-6, which is a PETISCO component. This interaction is mediated by a novel binding mode that does not involve the aromatic cage of TOFU-6's eTUD. They demonstrate that PUCH and PETISCO interact in vivo to process piRNA precursors and that piRNA cleavage occurs on the cytoplasmic surface of mitochondria by PUCH, which is brought there by the transmembrane domain of SLFL-3.

Overall this is a terrific paper. The genetic, biochemical, structural and enzymatic data are of very high quality and support the proposed model. The authors' findings are of broad interest and extend far beyond small RNA aficionados. They point to a deep convergence of mechanisms and localizations of piRNA biogenesis amongst diverse animals; and more generally of factors and strategies that organisms use to recognize and neutralize endogenous and exogenous selfish RNAs.

Author Rebuttals to Initial Comments:

We thank the reviewers for their time in evaluating our manuscript and for providing constructive comments. We address their comments below. *Original comments in italics*; replies in non-italics.

We hope that these revisions address the comments of the reviewers in a satisfactory manner.

Referee #1 (Remarks to the Author):

Major Comments:

1. *The authors show that PUCH strictly recognizes the m7G cap and make an intriguing proposal that this substrate specificity distinguishes piRNA precursors from mRNAs, which have trimethyl G (TMG) cap in C. elegans. Directly testing this model by using a TMG-capped AAU precursor in their cell free systems will significantly strengthen this study. Also, the authors should discuss if there can be any other m7G-capped transcripts in worms, and if so how PUCH (and PETISCO) acts on them.*

We tested a synthetic TMG-capped RNA, that had the same sequence as the m7G-capped RNA. We found that a TMG-capped RNA is not cleaved by recombinant PUCH (**Fig. 3f**), showing that Cap modification status indeed plays a role in substrate selection by PUCH. To exclude that the new TMG-capped RNA substrate may contain impurities that inhibit PUCH, we incubated our canonical Cap-AAU substrate in presence of access TMG-capped RNA. This did not affect PUCH cleavage (**Extended Data Fig. 6f**).

In addition to the new experiment shown in **Fig. 3f**, we also provide another experiment that addresses the same question. Here, we modified the m7G-capped RNA that we already had, using the enzyme TGS1. TGS1 is an enzyme known to tri-methylate m7G-capped RNA (Monecke, Dickmanns and Ficner (2009) Nucl. Acids Res. 37:3865-3877). We purified the TGS1 enzyme, treated our m7G-capped substrate, and ran PUCH cleavage reactions. As can be seen in **Fig. R1**: TGS1 treatment inhibited PUCH cleavage in a SAM-dependent manner. In order to save space in the manuscript, we decided not to include this particular experiment, but we wanted to show this as additional evidence in this rebuttal.

Regarding discussing other m7G capped transcripts in the worm: we now address this in the discussion in relation to the so-called type 2 piRNAs, which are derived from the 5' ends of dispersed, protein-coding genes (Gu et al. 2012). We now also show that these type 2 piRNAs are also produced by PUCH.

2. *The conserved "YR" motif is known to be important for transcription of piRNA precursors in worms. Do these two nucleotides (right upstream of the third "U") affect the processing by PUCH? Also, does PUCH have any preference for the length of substrate RNAs?*

The 'Y' of this motif is actually not part of the precursor RNA. The first nucleotide of the precursor is the 'R' (so A or G), making a 5' consensus sequence for precursors of RNU. We already showed that PUCH needs the U at position 3 for efficient cleavage. Testing the R at position -2 is indeed an

interesting question. To address this, we tested an m7G-capped substrate that starts with a 'C' in our *in vitro* processing reaction. We found that this 'CAU' substrate can still be cleaved by PUCH, but less efficiently than the AAU substrate (**Figure 3f** and **Extended Data Fig. 6d,e**). The new CAU substrate did not inhibit cleavage of the canonical AAU substrate (**Extended Data Fig. 6f**), excluding the possibility that the new substrate contained inhibiting impurities. We conclude that the R in the RNU consensus is not essential for, but does contribute to processing efficiency. To probe length aspects, we tested a substrate that is only 10 nucleotides long (shortened at the 3' end). This substrate was fully cleaved under our experimental conditions (**Fig. 3g**), indicating that length is not a critical determinant. We hope that with these novel additions the concerns of the reviewer are fully addressed.

3. As currently discussed in the manuscript, it will be important to experimentally examine the functional relevance of the transmembrane helix of SLFL-3 (and/or 4) and thereby the mitochondrial localization of PUCH in vivo.

To address this issue, we newly created the following set of mutants: *slfl-4* deletion, *slfl-3;slfl-4* double deletions and a *slfl-4* deletion;*slfl-3* deletion of transmembrane (TM) domain. Next, we sequenced mature and precursor piRNAs from these strains, and from the already presented *slfl-3* single deletion. We found that *slfl-3;slfl-4* double deletion mutants produce no piRNAs, while in single mutants piRNA levels are not significantly affected (**Fig. 2b**). Interestingly, deletion of the TM of SLFL-3, in absence of SLFL-4, also resulted in a strong loss of piRNAs, indicating that the TM domain is important for PUCH function (**Fig. 2b**).

4. In Fig. S1e, mature piRNAs levels appear to be comparable with their precursors in wildtype. In contrast, Fig. 5d and e show that overall mature piRNAs levels are >10 fold more abundant than their precursors in wildtype. Does this mean that the two piRNAs shown in Fig. S1e are peculiar examples?

We would like to start out mentioning that mature and precursor piRNAs cannot be reliably quantified from the same libraries because of the 5'-cap of piRNA-precursors which needs CIP/RppH de-capping treatment. Thus in the precursor libraries mature piRNAs are depleted, and *vice versa*. Therefore, mature piRNA plots cannot be directly compared to precursor plots, as they are based on different data-sets.

Fig. S1e shows normalized read coverage of two different piRNA loci in either wild-type or *tofu-2* mutant precursor piRNA libraries. Almost all reads start at the precursor position, consistent with the depletion of mature piRNAs. In the *tofu-2* mutant these reads pile up higher than in wild-type. These two loci are representative.

Fig. 5d and **e** (now panels **e** and **f**) showed mature and precursor libraries respectively. As mentioned, these cannot be directly compared. We have made these points more explicit in the figure legends.

5. Related to the above point, the current sequencing analysis relies on a very stringent definition of mature piRNAs and their precursors (pages 14–15), which may explain the apparent discrepancy above. The authors should present evidence to support the appropriateness of the definitions. Also, the authors should show how much fraction of non-structural reads are mapped to piRNA loci outside of mature piRNAs or piRNA precursors.

First, we want to emphasize that there is no discrepancy in the above data. We hope that our explanation provided in response to point 4 has made that clear.

The reason why we use a stringent definition of mature and precursor piRNAs is that we aimed for maximal specificity of the results, and to avoid mis-assignment of residual mature 21Us as piRNA precursors in the *precursor* libraries and *vice versa* in the *mature* libraries. We prefer to keep this like it is, as this manuscript is aimed at describing the nuclease that processes these precursor, and not at describing piRNA precursors per sé. Nevertheless, the reviewer is correct that perhaps with our new tools, we can extract novel properties, or refine precursor annotations. To test whether the stringent annotations affected our results, we tested our stringent quantification approach with a relaxed one in which we counted all 18-35nt reads that overlapped in sense orientation with piRNA (21ur) loci, in both *mature* (Fig. R2, a-c) and *precursor* libraries (Fig. R2, d-f).

Figure R2. Comparison of stringent versus relaxed mapping procedures in one of the data sets of the manuscript. ‘Stringent’: as described in the manuscript. ‘Relaxed’: all 18-35nt reads that overlapped in sense orientation with any piRNA (21ur) locus.

This revealed no substantial differences between the two approaches. As expected, more reads were counted in the ‘relaxed’ approach, but the overall results are the same. Basically, we find that the precursor descriptions are simply correct, and we hope that the reviewer agrees that there is no reason to expand on this topic in the current manuscript.

Minor Comments:

1. The statement “The identification of PUCH completes the repertoire of *C. elegans* piRNA biogenesis factors.” in Abstract may be too strong, as there may be other unknown piRNA biogenesis factors.

We have adjusted the sentence. It now reads:

“The identification of PUCH concludes the search for the 5’-end piRNA biogenesis factor in *C. elegans* and uncovers a novel type of RNA endonuclease formed by three SLFL proteins.”

2. Fig. 1d shows much more SLFL-3 than SLFL-4 in TOFU-2 IP, whereas Fig. S1d shows apparently no SLFL-3 in TOFU-2[E216A] IP. The authors should give a reasonable explanation for this.

SLFL-3 and SLFL-4 are very similar at the amino-acid level. Therefore, most peptides detected from these proteins are not unique for either SLFL-3 or SLFL-4. Within our initial analysis, we analyzed the raw MS spectra considering multi-mapping peptides using Occam’s razor principle (i.e. peptides mapping to more than one entry in the used database are added to the entry with the most unique peptides). This is a well-established procedure in MS-based proteomics and also the preset standard in the used MaxQuant analysis software. When we do the analysis only with unique peptides for

quantification, we find comparable levels of SLFL-3 and SLFL-4. We have replaced **Fig. 1d** with the corresponding volcano plot.

The IP-MS experiment for the E216A mutant of TOFU-2 unfortunately did not detect any unique peptides for SLFL-3 or SLFL-4. We have adjusted **Fig. S1d** and the legend to reflect this. Some words on the apparent loss of enrichment of SLFL-3/4 in the mutant IP-MS: enrichment values can vary significantly between independent experiments. Therefore, even if SLFL-3/4 is enriched less in **Fig. S1d** compared to **Fig. 1d**, this cannot be taken as evidence that the complex between the three proteins is less stable. We also refer to the new **Fig. S5h** where we assess the stability of recombinant wild-type or TOFU-2[E216A]-mutant PUCH: we detected no difference. Note also that we were able to purify the mutant complex over several columns, also indicating that the E216A mutation does not significantly affect PUCH stability. This is also mentioned in the legend to **Fig. S1d**.

3. In Fig. S1e, addition of vertical lines to indicate the 5' end of mature piRNAs will be helpful.

This has been added.

4. The overall upper shift in Fig. 1h etc. can be explained by the accumulation of piRNA precursors, the reduction of mature piRNAs, or both. An internal control will be helpful to distinguish these possibilities.

Because of the fact that mature piRNA are strongly depleted from precursor libraries, the effect we observe cannot be due to a loss of mature piRNAs. It is a relative increase in relation to any other type of RNA that is sequenced in these libraries. Due to the library preparation procedure, also miRNAs, 26G RNAs and 22G RNAs are depleted, leaving us no well-defined internal control we can use. However, given that we normalize against a complex mixture of RNAs, we believe that normalization is not a major problem for these plots. Also note that there is no accumulation of precursor RNAs in *tofu-2(e216a);pid-1* double mutants, whereas these mutants also do not have mature piRNAs (**Fig. 4a,b**). This indicates that the accumulation of precursors is not a quantification artefact resulting from the loss of mature piRNAs.

5. In Fig. 4b, does the shown p-value refer to tofu-2[E216A]-mutant or tofu-2[E216A];pid-1(xf35)?

In fact, the value relates to both samples, as basically both samples have close to zero mature piRNAs, resulting in the same P value in relation to the wild-type. This is now indicated in the legend.

6. Quantification and reproducibility should be shown in Fig. 4c.

This experiment (influence of PETISCO on PUCH activity) has been repeated two more times, and also with either IPed PUCH enzyme or mini-PUCH that was purified from *E. coli*. We provide both the gels and the quantifications of these experiments (**Fig. 4c,d**, and **Extended Data Fig. 6h,i**).

7. The right panel in Fig. 5f lacks "1" of "tofu-1." Also, tofu-1[L88R;L92R];tofu-6[V266E] may be better compared with tofu-6[V266E] rather than the wildtype.

This has been corrected. As for the comparison, we actually think contrasting the data to wild-type is better. The double mutant can of course also be compared to any of the two single mutants, but that would simply reduce the effect, as the single mutants also show minor shifts. This comparison can also be found in current **Fig. 5f**. The point of **Fig. 5g** is to show that the changes are found across all piRNA precursors.

8. "SLFL3/4" in Fig. S5a should read just "SLFL3."

This has been corrected.

9. Page 41, legend of Fig. S7d and e: “TOFU-1_72-182” and “TOFU-1_182” should probably read “TOFU-1_82-172” (as in the Figures).

This has been corrected.

10. After processing by PUCH, the resultant 5'-U piRNA precursors are expected to be loaded into PRG-1. Do the authors know if any components of PUCH interact with PRG-1?

We have not detected PRG-1 in any of our IP-MS experiments. It may interact with the eTudor domain of TOFU-6, but this will need to be tested. We added this to the discussion.

11. “Piwi” can mean a specific PIWI protein (among Aub, Ago3, and Piwi) in *Drosophila*. “PIWI” should instead be used when referring to the protein family to avoid any confusion.

This has been implemented.

Referee #2 (Remarks to the Author):

Based on these results, it could have been interesting to discuss more deeply the potential roles of SLFL proteins present in mammals.

Besides the viral SLFL protein we already mentioned, we now briefly also discuss the mammalian SLFNL1 protein, which also has only one SLFN fold. Unfortunately, not much is known about this protein, besides that its expression is enriched in testis. Any discussion that would go further than what we present would be too speculative in our opinion, and we prefer not to do that. We believe this should be addressed by future experiments before more meaningful discussions can be presented.

Major points to be addressed:

- Figure 2, panel a : Lanes 5 to 7 and to a lesser extent lane 4. The amount of eGFP-TOFU-2-WT used in these pull-down experiments differs compared to the other lanes. This is a problem as it is not possible to compare these lanes (which consist mostly in controls) with the other ones.

Indeed, TOFU-2 levels are lower whenever we do not co-express TOFU-1. Additionally expressing SLFL-3/4 leads to another raise in steady state levels of TOFU-2. We take this as a strong indication that TOFU-1, TOFU-2 and SLFL-3/4 proteins stabilize each other. However, the reviewer is right that this may affect some conclusions. We will specifically address each lane that was mentioned by the reviewer:

Lane 4: This lane tests whether TOFU-2 and TOFU-1 still interact in absence of SLFL-3/4. This is clearly the case. We would argue that the slightly lower expression of TOFU-2 is not an issue here: despite lower expression the two proteins still coIP.

Lane 5: This lane tests whether SLFL-3 is bound by TOFU-2 in absence of TOFU-1. Here we find that in absence of TOFU-1, TOFU-2 does not bind SLFL-3. This is indeed a case where lower TOFU-2 expression levels may affect the result. Depending on Kd's, SLFL-3 may be lost due to the lower concentration of TOFU-2. This would be in line with the structure predicted by AF2, where the interface between TOFU-1 and TOFU-2 is larger than between TOFU-2 and SLFL-3. Unfortunately, we cannot address this in this assay. We could provide more cell extract, but this will not raise the TOFU-2 concentration during the IP. Hence, we have added a note in the legend to reflect this limitation. We also observed, however, that SLFL-3 only affects TOFU-2 levels in presence of TOFU-1, supporting the idea that without TOFU-1, SLFL-3 does not significantly interact with TOFU-2. We mention this in the text.

The main point of this figure is to provide a first line of evidence that TOFU-1, TOFU-2 and SLFL-3 together form a complex. This aspect is further tested in the activity assays (**Fig. 3**), as well as by our recombinant protein of the trimer (**Fig. S5**). Hence, we believe that this caveat does not represent a major issue.

Lane 6: This lane shows TOFU-2 expression in absence of any co-transfected construct and demonstrates the specificity of all the antibody signals in the Western blots. No coIP is tested here and TOFU-2 levels are more or less irrelevant.

Lane 7: This lane tests whether mCherry alone displays affinity for TOFU-2. This is to control for the fact that TOFU-1 is tagged with mCherry in our experiment. The reviewer is correct that this conclusion can be affected by TOFU-2 expression levels. To control for this, we performed a coIP experiment where we enhanced TOFU-2 expression by means of TOFU-1 co-expression and tested whether TOFU-2 displays affinity for HA-mCherry. As shown in the anti-HA immunoblot in **Fig. R3**, this is not the case. We have not included this data into the manuscript, but we would be happy to include it, if this would be considered helpful.

Figure R3. TOFU-2 does not bind to mCherry. The indicated constructs were transfected into BmN4 cells, followed by IP of eGFP (=TOFU-2). We then probed for HA (=mCherry) to probe the interaction between TOFU-2 and mCherry, and FLAG (=TOFU-1). We also probed for FLAG (=TOFU-1) to confirm TOFU-1 and TOFU-2 interacted in this experiment. Note that TOFU-1 displayed some non-specific binding in this specific experiment, but the interaction is still clearly detectable. *: likely a band resulting from an eGFP-dimer, driven by the high expression levels of eGFP.

- Figure 3, panel b. In this experiment, endonuclease activity is only detected when the three wild-type proteins (TOFU-1, TOFU-2 and SLS-3) are present. The authors show a control with the TOFU-2 E216A mutant in the presence of TOFU-1. In this context, they do not detect enzymatic activity. However, as the complex between TOFU-1 and TOFU-2 WT does not exhibit enzymatic activity, this control cannot allow concluding that the TOFU-2 is the endonuclease and that E216 is a mandatory residue. A control of the TOFU-2 E216A mutant in the presence of both TOFU-1 and SLS-3 proteins is needed. Consequently, the conclusion « Introduction of an E216A mutation in TOFU-2 completely blocked this reaction (Fig. 3b). » (page 6, lanes 26-27) is not supported by the presented experiment.

Thank you for spotting this! We are afraid that this comment is based on errors in figure annotation. The E216E mutation was actually tested in context of all three PUCH subunits. Moreover, the recombinant PUCH E216A mutant also contains all three proteins, purified by affinity, heparin and gel filtration and is inactive. Hence, we hope the reviewer agrees that the conclusion is in fact fully supported by the data. We have corrected these errors.

- For several figures, the indication of the components used in the corresponding experiments is confusing. For instance, Fig. 3c, lanes 4 and 5 both contain cap-AAC precursor and recombinant WT PUCH. Is this correct? The same is true for lanes 6 and 7 of the same panel. Fig S7b, lanes 3 and 4: MBP-TOFU-1-(82-373) is wrongly indicated to be present in these conditions.

These figures have been corrected.

- *Figure 3d, lane 3. The authors seem to have omitted to mention that the cap-AAU precursor is present in lanes 3 and 4. In lane 3, the intensity of the band corresponding to the expected cleaved RNA ligated to the 10 nucleotide adapter is weak and the bands corresponding to the cap-AAU precursor and the cleavage product have higher intensity. This should be improved for better readability.*

We have improved the ligation efficiency. The revised **Fig. 3d** shows the new data. We have also corrected the figure annotation.

- *Figure 5e. It is crucial to validate that the different TOFU-1 and TOFU-6 mutants are well expressed in the cells used for these experiments. Without this control, it is not possible to conclude that the observed effect on levels of mature and precursor piRNAs are indeed due to the disruption of the interaction between TOFU-1 and TOFU-6.*

We used Western blotting on epitope-tagged TOFU-1 and TOFU-6 to probe how the missense mutations may affect TOFU-1 and TOFU-6. Both TOFU-1 and TOFU-6 were unaffected by the introduced mutations. These results are shown in **Fig. 5d**.

Minor points to be addressed :

- *The TOFU-2 E216A mutant seems to be less soluble than wild-type (Fig. S5g) and to interact more weakly with TOFU-1 and SLFL-3 (Fig. S1d). This should be discussed and in particular in light of the location of this residue in the model of the PUCH complex.*

First, we kindly refer to point 2 of reviewer 1 for the comment on **Fig. S1d**. Second, **Fig. S5f** and **g** depict peak fractions obtained from size-exclusion chromatography, and the varying band intensity on the gel simply arise from different quantities loaded per gel. Third, we have addressed the stability of recombinant PUCH in absence and presence of the TOFU-2[E216A] mutation. We utilized differential scanning fluorimetry to assess and compare the stability of the complexes (New **Fig. S5h**). In both cases, two unfolding transitions were observed. We do not know the molecular events that correspond to the two unfolding transitions; they may represent first complex dissociation followed by unfolding, or different subunits having varying degrees of stability. Nevertheless, the important result from this experiment is that no significant difference in stability between the two samples was observed: if anything, the E216A mutation appears even to shift the first melting point slightly towards a higher temperature. This strongly suggests that the catalytic mutation does not significantly impact the stability of the PUCH complex.

- *Page 6, lane 29. It is stated that the precursor processing activity is inhibited by EDTA. I have scrutinized the material and methods and all the text but I did not find any detail about EDTA concentration. It should be clearly indicated in Fig S6a-b, which lanes correspond to the experimental conditions with EDTA. I guess those labeled with a « - » sign but it should be clarified.*

This has been clarified in the methods. Briefly: BmN4 cells were lysed in EDTA+ Lysis Buffer (30mM Hepes [pH7.4], 150mM KOAc, 1mM EDTA and 0.1% Igepal freshly supplemented with EDTA-free protease inhibitor cocktail and 5mM DTT) after the IP, beads were washed with five times with EDTA+ Lysis Buffer followed by one wash in CB Buffer containing 1mM EDTA. After this, cleavage reactions were done in presence of the indicated concentrations of divalent cations (or in absence of divalent cations).

- Page 8, lanes 30-31 : It is not clear how the authors narrowed down the TOFU-1 region interacting with PETISCO from residues 82-172 to 82-113. This should be explained.

We added an explanation on how we derived the short TOFU-1 segment to the main text. Briefly, this was inspired by AlphaFold.

- Fig.1b. Please specify from which organism SLFN13 originates.

This has been added to the legend.

- Fig. S6b. It is stated in the materials and methods that the metal concentrations used in this experiment are 100 mM, which is radically different from the concentrations indicated on this panel. Please clarify.

This inconsistency has been corrected. The concentrations indicated on the panel were the correct values.

- Figure S9a : Due to the darkness, the details of this gel are difficult to see.

As the short TOFU-1 polypeptides are difficult to stain on polyacrylamide gels, we have repeated the experiments with GST-tagged TOFU-1 constructs. We have performed the pulldowns from both 'ends' by either using the MBP-tagged TOFU-6 eTUDOR domain or GST-tagged TOFU-1 constructs as bait. All constructs are now well visible on Coomassie brilliant blue stained polyacrylamide gels. This experiment is now shown in **Extended Data Fig.8a**.

- Figure S9c-e : The OD values for the Y-axes of these curves seem to have been normalized. From my point of view, this is quite confusing as one protein of the complex seems to be present in lower amount (differences in the height of the peak for TOFU-1 proteins alone or incubated with TOFU-6). I would suggest to use the measured OD values. This should facilitate the comparison between the different curves.

We have followed the advice of the referee and instead show the 'unnormalized' absorbance recordings. The changed figures are now shown in **Extended Data Fig. 8b-e**. To be consistent, we now show 'unnormalized' chromatograms throughout the manuscript and have updated **Fig.5b** and **Extended Data Fig. 7d,e** accordingly.

Typos to be corrected :

- Page 14 lane 15 and page 17 lane 8 : TRISOL should be TRIzol
- Page 14, lane 17 : 21xg seems to be too low. Please check.
- Page 19, lane 17 : Supdersex should be Superdex
- Page 19, lane 19 : Crystallisations should be crystallization.

We have corrected these typos.

E. Conclusions: robustness, validity, reliability

Some conclusions are not fully supported by the presented experiments.

As indicated above, this statement results from errors in figure annotation, for which we apologize. We hope the reviewer will now agree that the conclusions are fully supported by the data.

F. Suggested improvements: experiments, data for possible revision

In addition to the above mentioned major points, it would be interesting to indicate the location of E216 residues from TOFU-2 in the model of the TOFU-1/TOFU-2/SLFL-3 structure presented in Figure 2c.

We have added the three active site residues to the figure as suggested.

Referee #3 (Remarks to the Author):

We thank the reviewer for the extremely positive comments on our paper.

Reviewer Reports on the First Revision:

Referees' comments:

Referee #1 (Remarks to the Author):

The authors have addressed my previous concerns and the manuscript is now ready for publication.

Very minor points:

- Extend Data Fig. 1b is cited before Extend Data Fig. 1c-f.
- Extend Data Fig. 2f is cited before Extend Data Fig. 2d,e.
- Extend Data Fig. 9c is not cited.

Referee #2 (Remarks to the Author):

The authors have satisfactorily addressed all my comments.

Author Rebuttals to First Revision:

Rebuttal 2:

Referee #1 (Remarks to the Author):

The authors have addressed my previous concerns and the manuscript is now ready for publication.

Very minor points:

- Extend Data Fig. 1c-f is cited before Extend Data Fig. 1b.*
- Extend Data Fig. 2f is cited before Extend Data Fig. 2d,e.*
- Extend Data Fig. 9c is not cited.*

Referee #2 (Remarks to the Author):

The authors have satisfactorily addressed all my comments.

Answer to both reviewer 1 and 2:

We are glad the reviewers appreciated the revisions as appropriate and consider the manuscript ready for publication.

As identified by Reviewer 1: we corrected the order of citation of the relevant figure panels and now also cite Figure 9c.